# Decoupling Positional and Symbolic Attention Behavior in Transformers

**Felipe Urrutia**
CENIA & Faculty of Mathematics UC
Santiago, Chile
`felipe.urrutia@uc.cl`

**Jorge Salas**
CENIA
Santiago, Chile
`jorge.salas@cenia.cl`

**Alexander Kozachinskiy**
CENIA
Santiago, Chile
`alexander.kozachinskyi@cenia.cl`

**Cristian Buc Calderon**
CENIA
Santiago, Chile
`cristian.buc@cenia.cl`

**Hector Pasten**
CENIA & Faculty of Mathematics UC
Santiago, Chile
`hector.pasten@uc.cl`

**Cristobal Rojas**
IMC UC & CENIA
Santiago, Chile
`luis.rojas@uc.cl`

## Abstract

An important aspect subtending language understanding and production is the ability to independently encode positional and symbolic information of the words within a sentence. In Transformers, positional information is typically encoded using Positional Encodings (PEs). One such popular PE, namely Rotary PE (RoPE), has been widely used due to its empirical success. Recently, it has been argued that part of RoPE's success emerges from its ability to encode robust positional and semantic information using large and small frequencies, respectively. In this work, we perform a deeper dive into the positional versus symbolic dichotomy of attention heads behavior, both at the theoretical and empirical level. We provide general definitions of what it means for a head to behave positionally or symbolically, prove that these are two mutually exclusive behaviors and develop a metric to quantify them. We apply our framework to analyze Transformer-based LLMs using RoPE and find that all heads exhibit a strong correspondence between behavior and frequency use. Finally, we introduce canonical tasks designed to be either purely positional or symbolic, and demonstrate that the Transformer performance causally relates to the ability of attention heads to leverage the appropriate frequencies. In particular, we show that we can control the Transformer performance by controlling which frequencies the attention heads can access. Altogether, our work provides a detailed understanding of RoPE, and how its properties relate to model behavior.

## 1 Introduction

Positional Encodings (PEs) play a critical role in how modern Transformer-based Large Language Models (LLMs) function. Such criticality is, for example, illustrated by the fact that encoder-only Transformers with No Positional Encoding (NoPE) are *invariant* under permutations of prompt's tokens (Pérez et al., 2021), severely limiting their expressivity. While decoder-only Transformers using NoPE can in theory recover positional information (Kazemnejad et al., 2023), they are unable to learn some matrices observed in practice (Barbero et al., 2024). Regardless, at the practical scale, adding a component to explicitly inject positional information is the current standard choice in state-of-the-art Transformer-based LLMs.

Rotary PE (RoPE) has increasingly become one of the most popular choices to inject positional information. These PEs are given by a sequence $\theta_1, \ldots, \theta_{d/2}$ of $d/2$ angles (where the embedding

dimension $d$ is assumed to be even), each of which acts as a corresponding rotation on a different rotational plane. By rotating query and key vector pairs, they account for the relative positional distance between them, thereby incorporating the positional information into the computed representations. Conventionally, RoPE is believed to be helpful because it facilitates token dependency decay with increasing distance. Yet, more recent evidence suggests that RoPE's utility lies in its ability to leverage different frequency bands for different purposes. In Barbero et al. (2024) for instance, the authors show that while certain models largely prefer to use low frequencies (which they conjecture to implement "information channels"), they also find that specific particular heads use high frequencies to produce "robust positional attention patterns" that also play a key role in the overall workings of the model.

Moreover, the choice of the base in RoPE (which determines the total frequency range) has also been shown to play an important role when extrapolating models to longer contexts. On the one hand, there is evidence that improved performance in long-context fine-tuning can be achieved by decreasing the base (i.e., increasing the frequencies) in RoPE (Liu et al., 2023), a phenomenon often attributed to the fact that higher frequencies bias the Transformer towards *attending more to closer tokens*. On the other hand, decreasing the base negatively affects the ability to accurately retrieve information in such long contexts (Men et al., 2024). Indeed, information retrieval (in long contexts) requires attending to similar tokens *even when their relative distance is large*, an ability suggested to be improved by selecting larger (rather than smaller) bases (Men et al., 2024), thereby shifting the range towards lower frequencies.

At a conceptual level, all these works seem to point towards the fact that different frequency bands in RoPE are used by attention heads to implement two opposite core abilities: one more concerned with positional information and one with symbolic information, creating a tension between the two. Achieving a deeper understanding of this tension is a crucial step towards improving the mechanisms of Transformers by optimally striking the positional-symbolic balance in a given task context.

In particular, the following questions remain open:

- What are the mathematical properties that underlie these two core abilities?
- How can we measure if an attention head's behavior is exhibiting either of these abilities?
- How do these abilities relate to the use of the different frequencies in RoPE?
- How is model performance affected by the choice of these frequencies?

**Contributions.** In this work, we perform a detailed theoretical and empirical[1] study of all these questions. Due to space constrains, we defer our further related work discussion to Appendix A. We rely on the GEMMA-2, QWEN-2, and LLAMA-3 families of large language models for our empirical analysis. In summary:

1. We provide formal definitions of what it means for an arbitrary attention head to behave *purely positionally* or *purely symbolically* on a given input. We prove that these behaviors are mutually exclusive, unless the attention pattern follows a uniform distribution. Moreover, we prove that certain intrinsically positional operations cannot be implemented by heads that behave symbolically, whereas certain symbolic operations cannot be implemented by heads that behave positionally.

2. We introduce a novel metric intended to provide *scores* that reflect the extent to which a given head behaves positionally or symbolically on a given input. Our metric can be flexibly applied to analyze a given model at different levels of granularity, from specific heads on specific inputs and at specific frequencies in the case of RoPE-based attention, to a single pair of scores per head, enabling the characterization and visualization of the *positional-symbolic profile* of the model.

3. We apply our metric to inspect the behavior of attention heads in Transformer-based LLMs using RoPE, specifically the GEMMA-2, QWEN-2, and LLAMA-3 families. By analyzing these scores along the different frequency ranges, we reveal a surprisingly sharp connection between frequencies and attention head behavior: lower frequencies are preferred by symbolic heads, while moderately larger frequencies tend to correspond to positional heads. On

---

[1]Code for our experiments can be found at ⬤ furrutiav/positional-and-symbolic-iclr2026.

the other hand, larger frequencies are used by heads achieving high values in both scores, a phenomenon that forces an approximately uniform distribution of attention weights.

4. We then study attention heads in order to understand how frequency choices affect their performance in canonical tasks designed to intrinsically be either positional or symbolic. We find that a head can act positionally (and thus perform well on the respective task) only when given access to relatively large frequencies, while forcing it to work with lower frequencies inevitably leads to poor performance. We show that in this case, a $U$-shaped accuracy pattern tends to emerge. Similarly, attention heads can perform well on intrinsically symbolic tasks if given access to low frequencies. Interestingly, forcing it to rely on higher frequencies also harms performance, but this time leading to an *inverted $U$*-shaped accuracy pattern. Our observations are supported by both theoretical and empirical results.

## 2 PRELIMINARIES

**Definition 1** (Attention Head). *A $d$-dimensional decoder-only attention head is a function $H\colon (\mathbb{R}^d)^* \to (\mathbb{R}^d)^*$ given by a continuous "Logits function" $L\colon (\mathbb{R}^d \times \mathbb{N})^2 \to \mathbb{R}$, a continuous "Value function" $\mathrm{VAL}\colon \mathbb{R}^d \to \mathbb{R}^d$, and a continuous "activation function" $F\colon \mathbb{R}^d \times \mathbb{R}^d \to \mathbb{R}^d$. Given an input sequence of vectors $\bar{x} = (x_1, \ldots, x_n) \in (\mathbb{R}^d)^n$, the head $H$ outputs a sequence of vectors $\bar{y} = (y_1, \ldots, y_n) = H(\bar{x})$, computed as follows. First, the "attention weights" are computed as $w_j^i = \exp(\lambda_j^i)/\sum_{k \le i} \exp(\lambda_k^i)$, with $\lambda_j^i = L(x_i, i, x_j, j)$; then, a sequence $\bar{a} = (a_1, \ldots, a_n)$ of "attention vectors": $a_i = \sum_{j=1}^i w_j^i \cdot \mathrm{VAL}(x_j)$; and finally, one sets: $y_i = F(a_i, x_i)$, $i = 1, \ldots, n$.*

**Transformer architecture**. A (decoder-only) transformer $M_{\mathrm{EMB}}$ for the vocabulary $\Sigma$ is composed of an embedding function $\mathrm{EMB}\colon \Sigma \to \mathbb{R}^d$ where $d$ is the embedding dimension, a finite sequence of $d$-dimensional Attention Heads $H_1, \ldots, H_\ell$, and a function $r\colon \mathbb{R}^d \to \mathbb{R}^{|\Sigma|}$. Given a sequence $s = (\sigma_1, \ldots, \sigma_n)$ in the vocabulary $\Sigma$, the transformer $M$ returns a probability distribution on $\Sigma$ as follows: Define $x_i = \mathrm{EMB}(\sigma_i)$. Then, apply $H_\ell \circ \ldots \circ H_1$ to $(x_1, \ldots, x_n)$ to obtain $(u_1, \ldots, u_n)$. Subsequently, define $v = r(u_n) \in \mathbb{R}^{|\Sigma|}$ and finally the desired probability distribution is given by $\mu = \mathrm{softmax}(v)$. When $\mu$ is maximized at a unique symbol $\sigma \in \Sigma$ for an input $s$, we set $M_{\mathrm{EMB}}(s) = \sigma$.

**About the Logits function.** Of particular interest for us are the Logits functions of following form:
$$L(x_i, i, x_j, j) = \langle U^j K x_j, U^i Q x_i \rangle, \qquad i, j = 1, \ldots, n \quad j \le i, \tag{1}$$
where $Q$ and $K$ are the (learnable) "query" and "key" matrices and $U : \mathbb{R}^d \to \mathbb{R}^d$ is a unitary linear operator whose role is to incorporate the relative positional information between $x_i$ and $x_j$ given by positional inputs $i > 0$ and $j > 0$. We note that both NoPE and RoPE are special cases.

**Rotary Positional Encoding (RoPE)**. We assume an even hidden dimension $d \ge 2$. For RoPE (Su et al. (2024)) we consider the set of angles $G = (g_k = \theta^{-2(k-1)/d:k=1,\ldots,d/2})$ where $g_1$ is the fastest rotation angle/frequency and $g_{d/2}$ is the slowest one. For a given angle $g_k$, we denote the rotation matrix $\rho(g_k)$ as
$$\rho(g_k) = \begin{bmatrix} \cos(g_k) & -\sin(g_k) \\ \sin(g_k) & \cos(g_k) \end{bmatrix}.$$

To account for different rotation angles, we define a matrix $\mathbf{R}^i = \bigoplus_{k=1}^{d/2} \rho(g_k)^i \in \mathbb{R}^{d \times d}$ as the concatenation of the $i$-th exponentiation of rotation matrices $\rho(g_k)$ in a block diagonal matrix. Since each matrix $\rho(g_k)$ is a rotation matrix , we get that $\rho(g_k)^i = \rho(ig_k)$. Finally, for a query vector $x_i$ at position $i$ and key vector $x_j$ at position $j$, we define the RoPE logits function as
$$L_{\mathrm{RoPE}}(x_i, i, x_j, j) = (\mathbf{R}^j x_j)^\top (\mathbf{R}^i x_i) = x_j^\top \mathbf{R}^{i-j} x_i.$$

## 3 POSITIONAL VERSUS SYMBOLIC ATTENTION

As done in previous* transformer explainability (Ali et al., 2025; Sakarvadia et al., 2023; Ferrando et al., 2023; Ameisen et al., 2025) and RoPE analysis work (Barbero et al., 2024), and for the sake of simplicity, we choose to base our theoretical analyses directly on the Logits function, while attention weights are used for empirical observations. All proofs are deferred to the Appendix.

### 3.1 DEFINITIONS

In this section, we provide formal definitions of what it means for a head to behave positionally or symbolically on a given input. In the following, $S_i$ will denote the set of permutations of $[i]$.

**Definition 2** (Positional and Symbolic attention). *We say that a head $H$ acts **positionally** on an input $\overline{x} = (x_1, x_2, ..., x_n)$ at query $i \leq n$ if its logit sequence is* invariant *under permutation of the key vectors. That is, if for all $\pi \in S_{i-1}$:*

$$L(x_i, i, x_{\pi(j)}, j) = L(x_i, i, x_j, j) \qquad \forall j < i.$$

*On the other hand, we will say that $H$ acts **symbolically** on $\overline{x} = (x_1, x_2, ..., x_n) \in (\mathbb{R}^d)^n$ at query $i \leq n$ if its logit sequence is* equivariant *under permutations of the key vectors. That is, if*

$$L(x_i, i, x_j, \pi(j)) = L(x_i, i, x_j, j) \qquad \forall \pi \in S_{i-1}, \quad \forall j < i.$$

In other words, suppose we are querying on a given input from position $i$ to all positions $j \leq i$. A head $H$ is positional when $L(x_i, i, x_j, j)$ depends only on the position $j$, and not on the value of the key vector $x_j$. In contrast, $H$ will be symbolic when $L(x_i, i, x_j, j)$ only depends on the key-vector $x_j$ (the "symbol"), regardless of the position $j$. See Figure 4 for an illustration (Appendix B.1).

Let us mention a few important examples. First, it is easy to see that a NoPE head acts symbolically on every input (the logits function does not depend on $j$ in this case). Second, if an input $\overline{x}$ is such that the key vectors $Kx_j$ are all the same for $j = 1 \dots i - 1$, say equal to some vector $v$, then any head with Logits as in equation 1 will act positionally on it, simply because $L(x_i, i, x_j, j) = \langle v, U^{i-j}Qx_i \rangle$, which only depends on $j$. Note that for such an input, a NoPE head will act both positionally and symbolically. In this case, the logits $L(x_i, i, x_j, j)$ will have the same value for all $x_j$ with $j < i$, yielding a *uniform* attention pattern.

### 3.2 THE POSITIONAL-SYMBOLIC EXCLUSION PRINCIPLE

Let $L$ be a logit function and $\overline{x} = (x_1, ..., x_n)$ an input. Here, we assume we are querying from $i = n$. For a permutation $\pi \in S_{n-1}$ and $j \in [n-1]$, define

$$\delta_{L,\overline{x}}^{\mathrm{pos}}(\pi, j) = L(x_n, n, x_{\pi(j)}, j) - L(x_n, n, x_j, j),$$
$$\delta_{L,\overline{x}}^{\mathrm{sym}}(\pi, j) = L(x_n, n, x_j, \pi(j)) - L(x_n, n, x_j, j).$$

We view $\delta_{L,\overline{x}}^{\mathrm{pos}}$ and $\delta_{L,\overline{x}}^{\mathrm{sym}}$ as real $((n-1)! \cdot (n-1))$-dimensional vectors with coordinates, indexed by elements of $S_{n-1} \times [n-1]$. By definition, a head acts positionally (respectively, symbolically) at $i = n$ on the input $\overline{x}$ if and only if $\delta_{L,\overline{x}}^{\mathrm{pos}} = 0$ (respectively, $\delta_{L,\overline{x}}^{\mathrm{sym}} = 0$). One could imagine that some attention heads in real-life transformers, although not being exactly positional or symbolic, have one of the vectors $\delta_{L,\overline{x}}^{\mathrm{pos}}$ or $\delta_{L,\overline{x}}^{sym}$ really close to $0$, thus exhibiting "almost" positional or symbolic behavior. Based on this idea, in the next section we define *positional and symbolic scores*.

We now use these norms in a result that intuitively states the following: If the head's behavior on $\overline{x}$ is close to both positional and symbolic, then the attention weights form an approximately constant sequence. This is an instance of the *positional-symbolic duality*: only at the cost of a uniform non-focused attention, we can have both positional and symbolic behavior.

**Theorem 1** (The positional-symbolic exclusion principle). *Let $H$ be an arbitrary attention head with the logit function $L$. Let $\overline{x} = (x_1, ..., x_n)$ be an input and let $\lambda = (\lambda_1, ..., \lambda_{n-1})$ be the sequence of logits on this input, excluding $L(x_n, n, x_n, n)$; namely, $\lambda_j = L(x_n, n, x_j, j)$. Then, denoting by $\mu$ the average value of $\lambda_j$'s, we have*

$$\mathrm{Var}(\lambda) = \frac{1}{n-1} \sum_j (\lambda_j - \mu)^2 \leq \frac{\|\delta_{L,\overline{x}}^{\mathrm{pos}}\|_2^2 + \|\delta_{L,\overline{x}}^{\mathrm{sym}}\|_2^2}{(n-1)! \cdot (n-1)}$$

### 3.3 POSITIONAL AND SYMBOLIC SCORES AND THEIR APPLICATION TO REALISTIC MODELS

In this section, we introduce scores to measure the extent to which each head in a model behaves positionally or symbolically, and apply them to inspect the `google/gemma-2-2b-it` model (similar analyses on four other models are presented in the Appendix B.5). Our evaluation is grounded in

the *binding task*, which tests whether the model can correctly associate entities with their attributes (Feng & Steinhardt, 2023). Specifically, we construct inputs consisting of $n = 256$ entity–attribute pairs, where entities are proper names and attributes are colors (e.g., *Alice likes the color Red; Bob likes the color Blue*). At the end of the sequence, a query probes the attribute of one entity (e.g., *What color does Alice like the most?*). We choose this task for two reasons. First, it can intuitively be approached by a combination of positional and symbolic mechanisms. Second, we have control over the location at which a given entity-attribute pair appears in the prompt. This allows for a clearer comparison of the model behavior's when the pair location ranges from the beginning to the end of the prompt.

**From Logits to Attention weights.** Given an arbitrary head $H$ with logits function $L$ and an input $\bar{x}$, we define the logits produced when querying the final token $x_n$ as $L(\overline{x}) = \big(L(x_n, n, x_j, j) : j \leq n\big)$. Then, we define the corresponding attention weights as $D(\overline{x}) = \text{softmax}(L(\overline{x}))$.

**Computation of Positional and Symbolic Scores.** We partition the sequence $\overline{x}$ into contiguous $m$ blocks and compute the sequence $d = (d_1, ..., d_m)$, where $d_i$ is the average attention weight of $D(\overline{x})$ over the block $i$. To analyze how attention behaves under input permutations, we focus on simple block swaps (e.g., exchanging block $i$ with block $j$). For each such permutation, we represent the corresponding block averages before and after the swap into two-dimensional vectors $v_{ij} = (d_i, d_j)$ (before the swap) and $v'_{ij} = (d'_i, d'_j)$ (after the swap, but on the same block locations). Using cosine similarity on the two-dimensional vectors $v$ and $v'$, we define two scores: the positional score $s_{\text{POS}}(\bar{x})$, which measures the stability of block averages under permutations (i.e, we compare each $v'_{ij}$ against $v_{ij}$), and the symbolic score $s_{\text{SYM}}(\bar{x})$, which measures whether block averages move consistently with the permutation (i.e. we compare each $v'_{ij}$ against $v_{ji}$). The full details of how the metric is computed can be found in Appendix B.3.

**Visualizing heads in the Positional-Symbolic Plane.** The pair of scores $PS(H, \bar{x}) := (s_{\text{POS}}(\bar{x}), s_{\text{SYM}}(\bar{x}))$ defines a point in what we call the *positional-symbolic plane*. In order to visualize the behavior of an entire model on a given task, we can fix a prompt $\bar{\sigma} = (\sigma_1, \ldots, \sigma_n)$ and compute the scores for all heads in the model based on their corresponding inputs $\bar{x} = (y_1^{\ell-1}, \ldots, y_n^{\ell-1})$, with $\ell$ defining the layer number. This allows us to reveal the behavior of the entire model and visualize its *behavior's profile* in the *positional-symbolic plane*.

Figure 1 shows this profile for the entire `gemma-2-2b-it` model on the binding task. Each dot on Figure 1A represents the location of an attention head on the positional-symbolic plane (with color coding layer depth). We observe that early layer heads are associated to high positional scores, whereas late layer heads tend to be more symbolic[2]. Figure 1B, illustrates an important feature of our work. Compared to previous studies (Barbero et al., 2024), we now can depict a full model attention head behavior snapshot, for all heads and layers. This snapshot provides a strong empirical support for our theoretical positional-symbolic exclusion principle. Indeed, the scores in both heatmaps are negatively correlated (Pearson correlation coefficient: $r = -0.91$, p-value $\leq 0.0001$). Furthermore, Figure 7 shows the predominance of symbolic head behavior when querying block 1, and how the positional and symbolic scores gradually revert when the queried blocks gradually move from block 1 to 256.

**Positional and Symbolic Scores in RoPE.** We now take a closer look into how attention heads use the different RoPE frequencies. We do this by first decomposing a given head $H$ into $m$ distinct *projection heads*, each of which can be seen as an individual 2-dimensional head with a single associated frequency (see Appendix B.3 for details of logits per frequency). By computing the positional and symbolic scores for each projected head, we can visualize how the behavior of the combined head $H$ depends on each of the frequencies in RoPE. As can be seen in Figure 1C, there is a sharp correspondence between attention head behavior type (i.e., symbolic or positional) and frequency, with relatively high frequencies associated to positional behavior and lower frequencies to symbolic behavior, mirroring the results observed in some specific heads by Barbero et al. (2024), but now

---

[2]To assess the statistical significance of this pattern, we conducted a median test comparing early (layers 1-13) and late (layers 14–26) layers. For positional scores, early layers showed a higher median (0.83) than late layers (0.56; p-value $\leq 0.0001$). Conversely, symbolic scores were higher in late layers (median 0.67) than in early layers (median 0.49; p-value $\leq 0.01$).

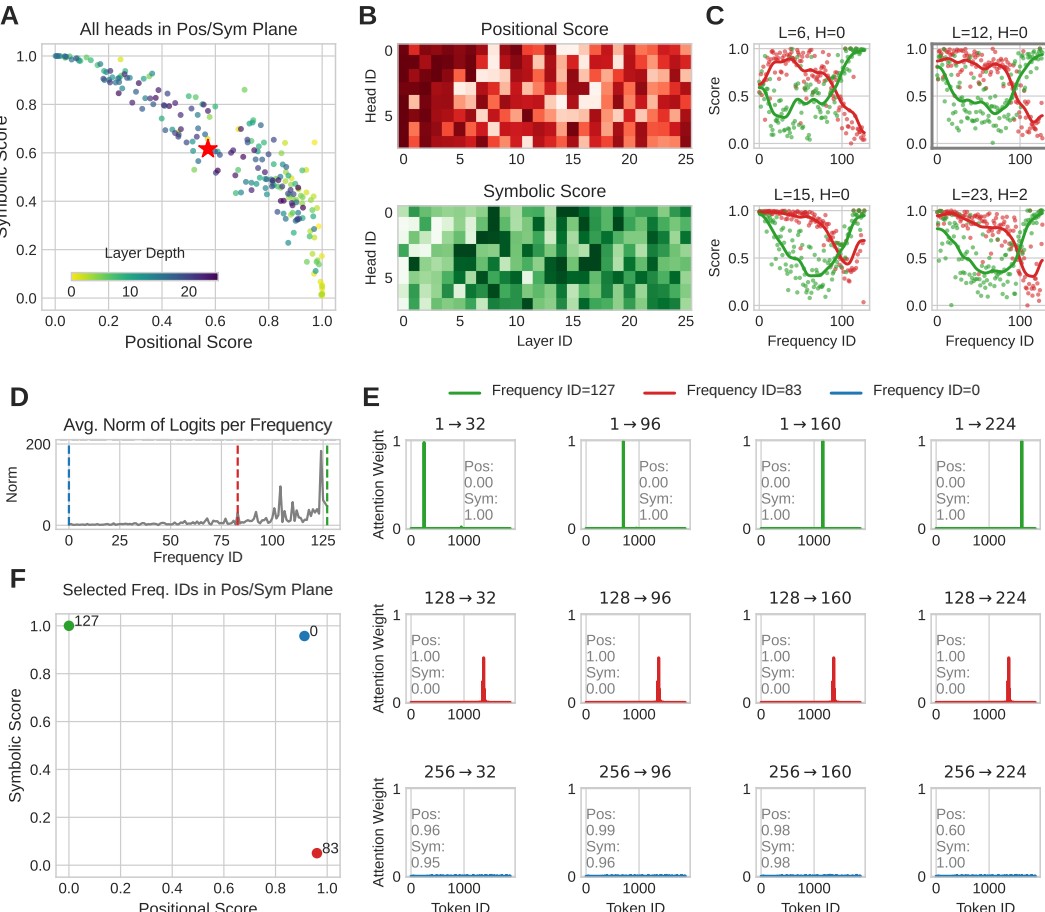

Figure 1: **Global and local analysis of attention head behavior. A.** Each head in the positional–symbolic plane. **B.** Heatmaps of positional and symbolic scores for each head across all layers. **C.** For the same heads, we plot their positional and symbolic scores as a function of RoPE frequencies. By convention, lower frequency IDs correspond to higher angular frequencies, and conversely, higher frequency IDs correspond to lower angular frequencies. **D.** norms of the logits at each frequency for head (12:0). **E.** Attention weight patterns as a function of permutations. Top rows (**green** traces): Behavior of a symbolic head whose attention weight mass follows the permutations. Middle rows (**red** traces): Behavior of a positional head whose attention weight mass is invariant to the permutations. Bottom rows (**blue** traces): Behavior of a (mix) attention head with both high symbolic and positional scores, displaying a uniform mass with low attention weight scores. Note that symbolic, positional, and mix head behavior ar associated with low, relatively large, and the largest frequencies, respectively. **F.** Location on the positional-symbolic plane of head (12:0) as a function of the selected frequency.

generalized to all heads and layers (see Figure 5). Our method allows to reveal a new empirical observation. We find that for the highest frequencies, some heads display large positional **and** symbolic scores. According to Theorem 1, these should correspond to heads exhibiting a uniform attention pattern.

We then took an even deeper look at the first head from layer 12 (head 12:0; see top right of Figure 1C). Figure 1E depicts the behavior of this head for three different frequencies exhibiting symbolic (**green** traces), positional (**red** traces) and both (**blue** traces) behaviors, respectively. As expected from our definitions, symbolic behavior is reflected by changes in the attention weight mass that moves with the permutation, whereas the attention weight profile remains unchanged for the positional behavior. One can also clearly see a uniform attention pattern for the frequencies where the head displays both behaviors. Finally, head (12:0) also allows us to understand the underlying

reasons for the location of heads on the positional-symbolic plane, as this head exhibits both high symbolic and positional scores (see red star on Figure 1A ). For this head, we analyzed the norms of the key vectors at each frequency (Figure 1D), and noted that most of the norm mass is concentrated on the smallest frequencies, which therefore contribute more to the global head's behavior, and explains a relative high symbolic score. Additionally, some mass is distributed over a large range of intermediate frequencies, in turn explaining a relatively high positional score.

## 4 POSITIONAL AND SYMBOLIC CANONICAL TASKS

In this section[3], we aim at deepening our theoretical understanding of positional and symbolic heads. In particular, we introduce non-trivial tasks that can be solved by pure positional (symbolic) heads using a 1–layer attention-only transformer. Our goal is to show how simple models can give us important clues with respect to the fundamental tension between tasks requiring the extraction of either positional or symbolic information from the input.

**Index Task.** We now introduce a task intended to be intrinsically positional. Consider an input sequence $s_{\text{POS}} = (\sigma_0, \sigma_1, \ldots, \sigma_{n-1}, j)$, where each $\sigma_i$ is a symbol from a finite vocabulary $\Sigma$, and $j$ is an integer index in the range $0 \leq j < n$. The goal of the task is to output the symbol at position $j$ (determined by the last token in $s_{\text{POS}}$), $\sigma_j$. Thus, to solve this task, one needs a transformer that computes the function $f_{\text{POS}}$ defined by $f_{\text{POS}}((\sigma_0, \sigma_1, \ldots, \sigma_{n-1}, j)) = \sigma_j$. Intuitively, any mechanism allowing to solve this task should be highly sensitive to the position being asked for, while it can ignore the actual symbols. We now proceed to prove that $f_{\text{POS}}$ can never be solved by a 1-layer decoder-only transformer $M_{\text{EMB}}$ with a single head $H$ that acts symbolically.

**Theorem 2.** *For an arbitrary attention head $H$, if $H$ acts symbolically on a non constant input $\bar{x}$ then $M_{\text{EMB}} \neq f_{\text{POS}}$.*

Note that Theorem 2 does not assume any specific form in which the positional information is incorporated into the logit function. On the other hand, our next result shows that $f_{\text{POS}}$ can be solved by a simple and purely positional attention head.

**Theorem 3.** *For each $n > 0$ there is a purely positional attention head $H_{\text{POS}}$ that uses RoPE with a single angle and no value or activation function, such that $M_{\text{EMB}} = f_{\text{POS}}$ for every $s_{\text{POS}}$ of length $n$.*

Even though solving $f_{\text{POS}}$ seems rather simple, Theorems 2 and 3 characterize it as a purely positional task. Interestingly, not every angle of RoPE is appropriate for the construction of $H_{\text{POS}}$. In fact, if the angle is greater than $2\pi/n$, attention weights are not maximized in just one position of the input, which could lead to confusion on the model's output. Later we will see how this effect emerges also empirically, having a direct impact on a model's capacity for solving tasks like $f_{\text{POS}}$.

**Information retrieval Task.** Here, we consider inputs of the form $s_{\text{SYM}} = (\sigma_1 \# i_1, \sigma_2 \# i_2, \ldots, \sigma_{n-1} \# i_{n-1}, \sigma_j \#)$ where $i_k$ is an integer for every $k < n$. The goal for this task is to retrieve the correct symbol $\sigma_j \# i_j$, associated to the last symbol $\sigma_j$ (the query). We assume for this task that $\sigma_j$ appears just once in the sequence before the query. Formally, the solution function $f_{\text{SYM}}$ is defined as $f_{\text{SYM}}((\sigma_0 \# i_0, \sigma_1 \# i_1, \ldots, \sigma_{n-1} \# i_{n-1}, \sigma_j \#)) = \sigma_j \# i_j$. Clearly, this task only depends on the symbols and positional information is irrelevant to the answer. Similar to the previous task, we can show that any simple head that acts positionally can not compute $f_{\text{SYM}}$. Here, by $H$ being simple we mean that VAL is the identity, the activation function $F$ is just a projection, and that EMB = ONEHOT.

**Theorem 4.** *Let $H$ be any simple attention head. If $H$ acts positionally on an input $\bar{x}$, then $M_{\text{ONEHOT}} \neq f_{\text{SYM}}$.*

We remark that if we allow $F$ to be a general MLP, then one can use it to override the attention mechanism and produce a (rather artificial) counterexample (see Appendix section C.6). We proceed by showing that a purely symbolic attention head $H$ can naturally solve $f_{\text{SYM}}$.

**Theorem 5.** *For every $n > 0$ there exists a purely symbolic attention head $H_{\text{SYM}}$, without value or activation function, such that $M_{\text{ONEHOT}} = f_{\text{SYM}}$ on every $s_{\text{SYM}}$ of size $n$.*

---

[3]All proofs for this section can be found in Appendix C.

**Experiments on positional versus symbolic heads.** Our results show that a simple 1-head transformer can perfectly solve $f_{\text{POS}}$ and $f_{\text{SYM}}$. Namely, the 1-RoPE $H_{\text{POS}}$ for $f_{\text{POS}}$ and the NoPE $H_{\text{SYM}}$ for $f_{\text{SYM}}$. In order to test if such heads can be learned during training and at the same time explore how the choice of the angle $\theta$ affects accuracy, we trained a simple 1-RoPE architecture for different values of $\theta$ on both tasks. While we observe that a 1-RoPE converges to a proper solution for the appropriate angles (Figure 2A), we also find that the chosen RoPE angle has a direct impact on the learning abilities of the model. Indeed, as expected from our theoretical results, small frequencies (large ID) are not suitable for solving the Index (positional) task, while larger ones are inadequate for the Information Retrieval (symbolic) one. Note the striking resemblance of these accuracy curves with the positional and symbolic scores curves from Figure 1C.

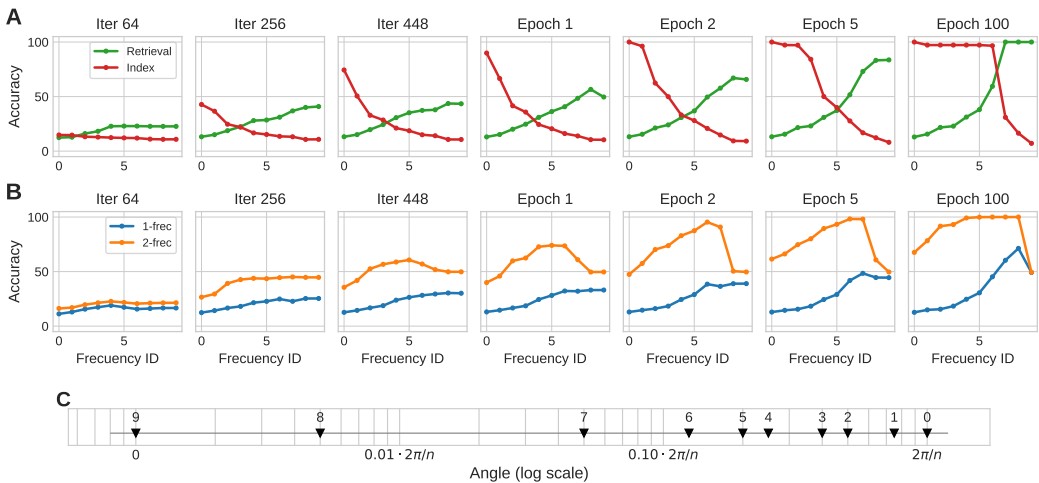

Figure 2: **Performance on the canonical tasks across training iterations and epochs. A.** Tension between Index (positional) and Retrieval (symbolic) tasks. **B.** Accuracy in the partial induction task for 1-frequency and 2-frequency models. **C.** Frequency IDs mapped to the log-scaled angle axis.

**Partial induction Task.** Finally, we present a partial induction task[4] with inputs of the form $s_{\text{MIX}} = (\sigma_1 \# i_1, \sigma_2 \# i_2, \ldots, \sigma_{n-1} \# i_{n-1}, \sigma_j \#)$ where $i_k$ is an integer for every $k < n$. We assume for this task that $\sigma_j$ appears at least two times in the sequence, in positions $j_1$ and $j_2$, and that integers $i_{j_1}$ and $i_{j_2}$ are different. The goal for this task is to retrieve the correct symbol-integer pair associated with the **last occurrence** of $\sigma_j$ (notice that the case with a single occurrence subsumes the information retrieval task). We denote by $f_{\text{MIX}}$ the corresponding solution function. The following result (which follows directly from our previous results) show that if a 1-layer transformer computes $f_{\text{MIX}}$, then it cannot behave in a purely positional or purely symbolical way.

**Corollary 1.** *If a simple model $M_{\text{ONEHOT}}$ with head $H$ computes $f_{\text{MIX}}$, then for every input $s_{\text{MIX}}$, $H$ does not behave positionally (or symbolically) on $s_{\text{MIX}}$.*

On the positive side, we demonstrate that a single attention head with two RoPE angles suffices.

**Theorem 6.** *For every $n > 0$ there exists an attention head $H_{\text{MIX}}^{0,\theta_2}$ with two RoPE angles such that $M_{\text{ONEHOT}} = f_{\text{MIX}}$ on every $s_{\text{MIX}}$ of length $n$.*

**Experiments for the partial induction task.** Figure 2B, shows that a toy model with 1-RoPE angle can not solve the partial induction task. On the other hand, a model with angles $\theta_1 = 0$ and $\theta_2$ (flexible), can solve it when $\theta_2$ is not too big. This is expected from the proof of Theorem 6 since bigger angles for $\theta_2$ confuse the model's output. Finally, the fact that the 1-RoPE version of this model has a decreasing accuracy as angles get bigger suggests that our version of the partial induction task has a more prominent symbolical difficulty. This could be explained by the fact that solving $f_{\text{MIX}}$ requires to detect only two positions of the input (both occurrences of $\sigma_j$), but several symbols need to be distinguished at the same time.

---

[4]Named as such in allusion to induction heads operations (Olsson et al., 2022).

## 5 ON THE RELATION BETWEEN FREQUENCIES AND ACCURACY SHAPE

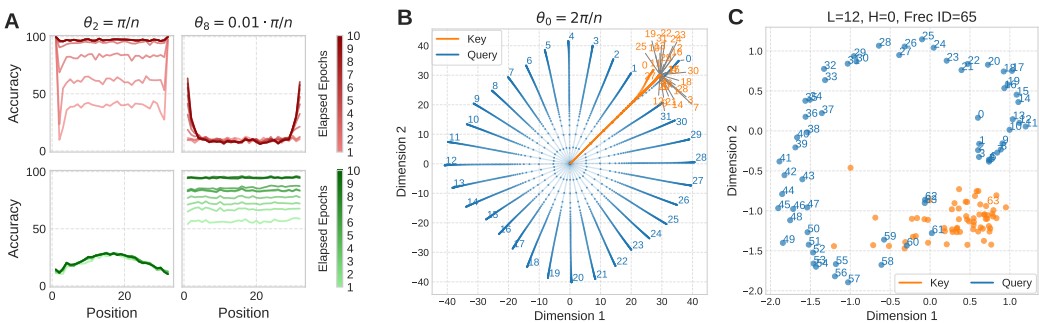

Figure 3: **A.** Shapes of Accuracy for the Index (in **red**) and Retrieving (in **green**) tasks, per Epochs. **B.** Query (in **blue**) and key (in **orange**) vectors trajectories during training for the Index Task. **C.** Query/Key vector projections on a rotational plane from a `gemma-2-2b-it`'s Head for some Binding Task's input ($64$ pairs).

As we have seen, a head required to behave in a certain way needs to access certain frequencies. In this section, we are interested in understanding how the performance of a head that is forced to operate with the wrong frequency is affected. We thus analyzed the accuracy of our trained 1-layer Transformer on both of our canonical tasks, focusing on how it depends on the location of the answer within the prompt. Interestingly, as it can clearly be seen in Figure 3A (upper graphs), the loss in accuracy when using frequencies that are too low for solving the Index task (**red**) is more severe when the answer lies at the middle of the prompt, and less severe when they closer to the border positions, resulting in a $U$-shaped accuracy pattern. This is reminiscent of a phenomenon that has been reported before as "lost in the middle" (Liu et al., 2024). On the other hand, when solving the Retrieving task (**green**) using frequencies that are not low enough, the opposite phenomenon is observed, and an *inverted $U$*-shaped accuracy pattern emerges.

Figure 3B shows the trajectories of the query (**blue**) and key (**orange**) vectors during training. These trajectories reveal the emerging mechanism the trained model relies on to solve the task. The query vectors are configured so that their angles code the different positions they are querying to, while the key vectors arrange into a single direction, which explains why the head behaves positionally. In fact, this is the exact same mechanism the theoretical solution $H_{\text{POS}}$ implements. On the other hand, Figure 3C shows the projections of the query (**blue**) and key (**orange**) vectors on a selected rotational plane for a frequency where head 12:0 from `gemma-2-2b-it` behaves positionally. One can see the striking resemblance to the toy model.

By further studying the mechanisms implemented by our theoretical toy models, we are able to provide an explanation for the observed accuracy shapes, at least at the attention level pattern. Let us say that a function $f : [n] \to \mathbb{R}$ is *u-shaped* if it has local maxima at $0$ and $n$ and over this interval it is first decreasing and then increasing. Similarly, we say it is *inverted-u-shaped* if $-f$ is *u-shaped*. Now, for either of our canonical tasks (index and information retrieval) let $j$ be the location of the answer within the input prompt, and let $w_{\max}(j)$ be the largest attention weight value of the heads $H_{\text{POS}}$ and $H_{\text{SYM}}$ from the proofs of Theorem 3 and 5, respectively. For the next Theorem, we consider a modified version $H_{\text{SYM}}^{\theta}$ of $H_{\text{SYM}}$ where we allow for the use of an angle $\theta$. From the construction of these heads it follows that $w_{\max}(j)$ is precisely the weight at position $j$, which is moreover equal for every such input. Besides our empirical findings, in the Appendix we give a rigorous mathematical proof of the following:

**Theorem 7.** *The function $w_{\max}$ is u-shaped for $H_{\text{POS}}$ and inverted-u-shaped for a simplified version of $H_{\text{SYM}}^{\theta}$.*

## 6 CONCLUSIONS AND FUTURE DIRECTIONS

We have carried a thorough analysis of two core abilities that attention heads in Transformers require in order to work effectively. By providing a precise definition of positional and symbolic behavior,

we came up with a metric to detect a working head's positional or symbolic nature. We were able to disentangle these mechanisms and mathematically understand why these two abilities are in tension with each other. By analyzing both real and toy models, we explained why heads prefer different frequencies in RoPE depending on the mechanisms they need to implement to process a given input, and finally related the resulting accuracy of a model to its ability to employ the appropriate frequencies.

We believe the characterization of a model by its positional-symbolic profile is a powerful tool to analyze how particular models behave on different tasks. Even though in this paper we have concentrated on the Binding Task (Feng & Steinhardt, 2023), we foresee a promising strategy in systematically performing similar analyses for a wider set of tasks and models. For example, one could use this tool to characterize the nature of models and tasks in terms of the "amount" of positional or symbolic mechanisms they tend to implement (for models) or rely on (for tasks).

In fact, the invariant properties that underlie positional or symbolic behavior can be thought of as inductive biases that the model has to learn to solve the task at hand. In other architectures such as CNNs or GNNs for example, these type of inductive biases are imposed into the architecture by design, thereby improving generalization properties and learning efficiency (Maron et al. (2019); Petrache & Trivedi (2023); Deng et al. (2022)). It is interesting to wonder whether a similar strategy could be applied to Transformers, for instance by imposing the appropriate degree of positional or symbolic behavior to selected heads. It is tempting to conjecture that these two core abilities constitute in fact the fundamental building blocks of any head, in the sense that their action can always be somehow decomposed into simpler heads behaving purely either positionally or symbolically. Such a theory could constitute the base for designing architectures targeted to specific tasks. We see this as an exciting avenue for future work.

## ACKNOWLEDGMENTS

All authors are funded by the National Center for Artificial Intelligence CENIA FB210017, Basal ANID. Kozachinskiy is supported by ANID Fondecyt Iniciación grant 11250060.

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

## A   RELATED WORK

Given the success of RoPE, several studies have dived into its mechanisms to provide: (*i*) better long-context performance, and (*ii*) a deeper knowledge of how the RoPE's mechanisms relate to the type of information encoded in the Transformer.

**Extending context windows.**   Much work has been done to modify or extend the mechanisms of RoPE in order to extend the context window length of LLMs. It has been shown down-scaling position indices on RoPE, within the pre-trained context-length limit and alongside small amounts of fine-tuning, avoids catastrophic attention scores blow-ups seen in naive extrapolation, thereby promoting extrapolation in unseen ranges (Chen et al., 2023b). In the same vein, Peng et al. (2023) propose YaRN, a method that combines Neural-Tangent-Kernel aware position interpolation with attention scaling to improve long context window performance. Other researchers have found that increasing the RoPE base frequency improves long-context information retrieval (Xiong et al., 2023). More recently, LongRoPE proposes the addition of non-uniform position interpolation with a fine-tuning curriculum strategy as shown to further increase the ability of long-context performance (up to 2,048k tokens) (Ding et al., 2024). Finally, by analyzing attention scores in the Needle In A Haystack [5] benchmark, Yang et al. (2025) observed that NoPE was related to higher attention mass over the needle (compared to RoPE), leading them to propose RNoPE, i.e., alternating NoPE and RoPE layers. RNoPE resulted in better performance, particularly within question-answering contexts .

**Mechanisms of RoPE and their relation to information encoding.**   Despite its importance to improve model performance, much less work has focused on generating a deeper theoretical knowledge of the mechanisms subtending RoPE, and how these relate to the behavior of attention heads, and in turn to model performance. Chen & Yan (2024) observed that the 2d pair rotation direction highly depends on Query/Key pair weight angle. These authors observe empirically, and theoretically demonstrate, that non-orthogonal weight vector pairs are more sensitive to positional information, while orthogonal ones focus rather on semantics. Their theoretical advances was then used to devise an efficient fine-tuning method (Angle-based Weight Masking) that primarily focuses on updated semantics-related weights. Closest to our work is that of Barbero et al. (2024). In their work, they theoretically prove (and empirically observe in Gemma 7B) that RoPE can learn specific positional (diagonal and off-diagonal) attention head patterns, which is not the case for NoPE. Furthermore, they relate the ability to learn these position attention patterns as leveraging high frequencies in RoPE. Conversely, semantic attention patterns are attributed to leveraging low frequencies. Furthermore, they suggest to truncate low-level frequencies to avoid arbitrary (irrational) rotation values in long contexts, and observe better performance when doing so (when keeping 75% of the highest RoPE frequencies). In works like Kitaev & Klein (2018) they hard wire the separation of positional and symbolic (or content) information present in the input, by forcing this separation in the initial embeddings. Ke et al. (2020), on the other hand, incorporates particular projection matrices for positions and symbolic information independently, which are then aggregated as two different components of the logits function. In this work, we do not force any separation of positional or symbolic information in the models. Rather, we relate these qualities to the behavior of the logits function, with a particular interest in the context of RoPE as the state-of-the-art positional encoding. We found that, in decoder-only Transformers, there is an emergence of heads tending to either adopt positional or symbolic behavior (which is learned and not forced in the architecture) suggesting that many underlying mechanisms can be decomposed into purely symbolic and purely positional ones. In this sense, our work provides strong theoretical and empirical reasons for the study of architectures with the capacity of learning separate positional and symbolic logit functions. Finally, in Chen et al. (2023a) researchers investigate attention patterns and defines interesting metrics of locality and symmetry of attention patterns in Transformers. In comparison, instead of considering attention weight patterns as they stand, our metric analyzes how these patterns change when permuting the input vectors, thus capturing mechanistic properties of a different nature. In particular, our metric is in principle agnostic to how focused these patterns are, and for instance both symbolic and positional behavior are perfectly compatible with a focused attention.

---

[5] https://github.com/gkamradt/LLMTest_NeedleInAHaystack

Our work extends current knowledge by: (*i*) mathematically defining positional or symbolic attention head behavior, (*ii*) offering a novel metric to quantify these behaviors at different levels of granularity (from heads to full model), (*iii*) theoretically proving that position and symbolic attention heads are mutually exclusive, (*iv*) theoretically proving that some positional tasks cannot be performed by symbolic heads, and vice-versa, and (*v*), empirically demonstrating that we can causally control the accuracy pattern in by gating access to specific frequencies.

Moreover, as described in the main text, we use a toy 1-head decoder-only transformer to gain a deeper understanding of the RoPE properties. Such a model allows us to have a strong hold on the interpretation of the RoPE mechanism, otherwise lost when working with deeper and wider models. Other works following this approach have demonstrated important features of Transformers. For instance, such toy models have been used to characterize transformer loss landscapes (Makkuva et al., 2024; 2025). Furthermore, 1-layer transformers have also been used to evaluate their expressivity Li et al. (2024b); Quirke & Barez (2023); Sanford et al. (2023; 2024); Kozachinskiy et al. (2025). Moreover, these models have been used to understand the learning dynamics and properties in transformers (Tian et al., 2023; Li et al., 2024a; Huang et al., 2024; Yang et al., 2024; Chen et al., 2024).

# B ADDITIONAL ILLUSTRATION AND PROOFS OF SECTION 3

## B.1 ILLUSTRATION OF ATTENTION HEAD BEHAVIOR DEFINITION

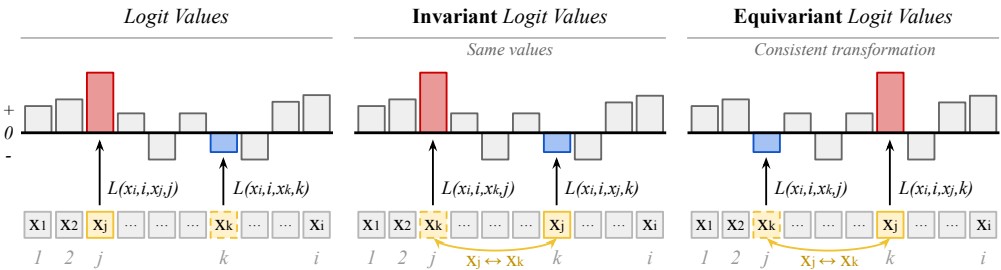

Figure 4: Illustration using the same sequence as in Definition 2. **Left:** Original logit values. **Middle:** Example of *invariant* logit values after a simple permutation (swapping $x_j$ with $x_k$). **Right:** Example of *equivariant* logit values under the same permutation.

## B.2 PROOF OF THEOREM 1

Denote $a_{k,j} = L(x_n, n, x_k, j)$. Take an index $j \in [n-1]$ and a permutation $\pi \in S_{n-1}$ independently and uniformly at random. Note that $(j, \pi(j))$ is a uniformly random pair from $[n-1]^2$ (under any fixation of $j$, the distribution of $\pi(j)$ is uniform). Observe that:

$$\mathrm{Var}(\lambda) = \mathrm{Var}(a_{jj}),$$

$$\frac{\|\delta_{L,\bar{x}}^{\mathrm{pos}}\|_2^2}{(n-1)! \cdot (n-1)} = \mathbb{E}(a_{\pi(j)j} - a_{jj})^2,$$

$$\frac{\|\delta_{L,\bar{x}}^{\mathrm{sym}}\|_2^2}{(n-1)! \cdot (n-1)} = \mathbb{E}(a_{j\pi(j)} - a_{jj})^2.$$

Since $(j, \pi(j))$ is a uniformly random pair from $[n-1]^2$, the pair $(\pi(j), j)$ has the same distribution, meaning that:

$$\frac{\|\delta_{L,\bar{x}}^{\mathrm{sym}}\|_2^2}{(n-1)! \cdot (n-1)} = \mathbb{E}(a_{\pi(j)j} - a_{\pi(j)\pi(j)})^2.$$

Using the fact that $\mathrm{Var}(X) = \mathbb{E}(X - Y)^2/2$ for any random variable $X$ and its independent copy $Y$, we get:

$$\mathrm{Var}(\lambda) = \mathrm{Var}(a_{jj}) = \mathbb{E}(a_{jj} - a_{\pi(j)\pi(j)})^2/2.$$

Finally, using an inequality $(x-y)^2 \leq 2x^2 + 2y^2$ for $x, y \in \mathbb{R}$, we derive the statement of the theorem

$$\mathrm{Var}(\lambda) = \mathbb{E}(a_{jj} - a_{\pi(j)\pi(j)})^2/2 \leq \mathbb{E}(a_{jj} - a_{\pi(j)j})^2 + \mathbb{E}(a_{\pi(j)j} - a_{\pi(j)\pi(j)})^2$$

$$= \frac{\|\delta_{L,\bar{x}}^{\mathrm{pos}}\|_2^2 + \|\delta_{L,\bar{x}}^{\mathrm{sym}}\|_2^2}{(n-1)! \cdot (n-1)}$$

## B.3 GENERAL METRIC TO TEST POSITIONAL OR SYMBOLIC HEADS

**Blocks.** We define a fixed partition of the index set $[n]$ into $m$ contiguous intervals (blocks)

$$I_1 = [a_1, b_1], \quad I_2 = [a_2, b_2], \quad \ldots, \quad I_m = [a_m, b_m], \qquad a_1 = 1, \ b_m = n,$$

with $a_{k+1} = b_k + 1$. For block $I_k$ define its average attention weights under $\bar{x}$ by

$$d_k(\bar{x}) = \frac{1}{|I_k|} \sum_{t \in I_k} D(\bar{x})_t.$$

We collect these into the block-average vector $d(\bar{x}) = (d_1(\bar{x}), \ldots, d_m(\bar{x})) \in \mathbb{R}^m$.

**Block permutations with dynamic intervals.** Let the original sequence be partitioned into contiguous blocks $I_1, \ldots, I_m$ as above, with lengths $|I_1|, \ldots, |I_m|$. A block permutation $\pi \in S_m$ rearranges these blocks, producing a new sequence $\pi(\overline{x})$. We then define new block intervals $I'_1, \ldots, I'_m$ by preserving the original *relative positions* but adjusting the boundaries according to the lengths of the permuted blocks. More precisely, we let

$$|I'_k| = |I_{\pi^{-1}(k)}|, \qquad I'_k = [a'_k, b'_k],$$

where the boundaries are given recursively by

$$a'_1 = 1, \qquad b'_k = a'_k + |I_{\pi^{-1}(k)}| - 1, \qquad a'_{k+1} = b'_k + 1.$$

Thus, the $k$-th block position in the permuted sequence has the same length as the block that was moved into that position. The block averages under the permuted sequence are

$$d_k\big(\pi(\overline{x})\big) \;=\; \frac{1}{|I'_k|} \sum_{t \in I'_k} D\big(\pi(\overline{x})\big)_t.$$

**Two–dimensional representation.** A simple type of permutation $\pi$ are a swap between block $i$ and block $j$. We represent the relevant attention mass by the vector

$$v_{ij}(\overline{x}) = (d_i(\overline{x}), \, d_j(\overline{x})),$$

and after the permutation by

$$v_{ij}(\pi(\overline{x})) = (d_i(\pi(\overline{x})), \, d_j(\pi(\overline{x}))).$$

**Permutation weights.** Not all block swaps are equally informative: we weight each permutation $\pi$ according to the amount of mass it moves,

$$\alpha(\pi) = \mathrm{softmax}\big(\big|d_i(\overline{x}) - d_j(\overline{x})\big|/\tau\big),$$

where $\tau > 0$ is a temperature parameter controlling how sharply the softmax focuses on high-mass swaps.

**Positional and symbolic scores.** We can now quantify how the head behaves with respect to block permutations using cosine similarity[6]. The *positional score* measures how stable the block averages remain under a permutation:

$$s_{\mathrm{POS}}(\overline{x}) = \sum_\pi \alpha(\pi) \, \cos\mathrm{sim}\Big(v_{ij}(\pi(\overline{x})), \, v_{ij}(\overline{x})\Big).$$

The *symbolic score* instead measures whether the block averages transform exactly as the permutation prescribes:

$$s_{\mathrm{SYM}}(\overline{x}) = \sum_\pi \alpha(\pi) \, \cos\mathrm{sim}\Big(v_{ij}(\pi(\overline{x})), \, v_{ji}(\overline{x})\Big).$$

**Scores per frequency.** Even though previous definitions are applicable to any logits function $L$, in the case of RoPE we can leverage its frequency decomposition (as a Hadamard attention head) to have a more refined understanding of the positional and symbolic behavior of each RoPE's frequency as single angle attention heads.

We begin by isolating the contribution of a single attention head at a given layer, and decomposing its logits into frequency components induced by RoPE. Consider layer $\ell$ and head $h$. The logits between query position $n$ with token $x_n$ and key position $j$ with token $x_j$ are given by

$$
\begin{aligned}
L^{\ell h}(x_n, n, x_j, j) &= \Big\langle K^{\ell h} x_j, \, R_\Theta^{n-j} Q^{\ell h} x_n \Big\rangle \\
&= \sum_{t=1}^{d/2} \Big\langle K_{2t, 2t+1}^{\ell h} x_j, \, R_{\theta_t}^{n-j} Q_{2t, 2t+1}^{\ell h} x_n \Big\rangle \\
&= \sum_{t=1}^{d/2} L_t^{\ell h}(x_n, n, x_j, j).
\end{aligned}
$$

---

[6] $\cos\mathrm{sim}(a, b) \;=\; \frac{\langle a, b\rangle}{\|a\|\,\|b\|}$.

Here each term

$$L_t^{\ell h}(x_n, n, x_j, j) = \left\langle K_{2t,2t+1}^{\ell h} x_j, \; R_{\theta_t}^{n-j} Q_{2t,2t+1}^{\ell h} x_n \right\rangle$$

corresponds to the logit contribution from frequency $\theta_t$ where the metrics $s_{\text{POS}}(\overline{x})$ or $s_{\text{SYM}}(\overline{x})$ can be measured individually.

## B.4 TEMPLATE PROMPTS FOR REAL-WORLD MODELS ON BINDING TASK

In our template prompts, we consider the span-block format defined as follows:

```
span_block = "\n".join([f"{name} likes the color {color}"
for name, color in sequence])
```

See bellow the template prompts for each model.

---

**gemma-2-it**

```
<bos><start_of_turn>user
You are a helpful assistant that can answer questions about entities and their associated
attribute.
---
Context:
{{span_block}}
---
Question: What color does {{entity[j]}} like the most?<end_of_turn>
<start_of_turn>model
Answer: {{entity[j]}} likes the color
```

---

**llama-3.2-it**

```
<|begin_of_text|><|start_header_id|>system<|end_header_id|>
You are a helpful assistant that can answer questions about entities and their associated
attribute.<|eot_id|>
<|start_header_id|>user<|eot_id|>
Context:
{{span_block}}
---
Question: What color does {{entity[j]}} like the most?<|eot_id|>
<|start_header_id|>assistant<|eot_id|>
Answer: {{entity[j]}} likes the color
```

---

**qwen2-it**

```
<|im_start|>system
You are a helpful assistant that can answer questions about entities and their associated
attribute.<|im_end|>
<|im_start|>user
Context:
{{span_block}}
---
Question: What color does {{entity[j]}} like the most?<|im_end|>
<|im_start|>assistant
Answer: {{entity[j]}} likes the color
```

---

## B.5 GEMMA-2-2B-INSTRUCT ON THE BINDING TASK

For the `google/gemma-2-2b-it` model, we compute global positional and symbolic scores for each attention head, and for each head we compute the scores of the projected heads for every frequency. The same analysis is also performed for other models, with results deferred to the next sections. Our evaluation is grounded in the *binding task*, which tests whether the model can correctly associate entities with their attributes (Feng & Steinhardt, 2023). Specifically, we construct inputs consisting of $n = 256$ entity–attribute pairs, where entities are proper names and attributes are colors (e.g., *Alice likes the color Red; Bob likes the color Blue*). At the end of the sequence, a query probes the attribute of one entity (e.g., *What color does Alice like the most?*). The choice of the task is made based on two reasons. First, it is a task that can intuitively be approached by a combination positional and a symbolic mechanisms. And second, we have complete control of the location of

the correct answer the question at the end of the prompt requires the model to find. This allows for a clearer comparison of the model behavior's when we change this location by changing the query.

We segment each input into 256 blocks, with each block containing exactly one entity–attribute pair. Then we select 9 uniformly spaced blocks. For a given query targeting the $k$-th block, we generate alternative inputs by permuting the position of $i$-th block with a different $j$-th block, for all $i, j = 1...9$. This procedure yields nine permutations per query, and a total of nine queries per input.

For each attention head, we compute the positional and symbolic scores across RoPE frequencies. Recall that Gemma employs RoPE, where frequency identifiers are mapped to angular frequencies according to $\theta_t = 10000^{-2 \cdot t/d}$, with $t$ the frequency index and $d = 256$ the head dimension. By convention, lower frequency IDs correspond to higher angular frequencies, and conversely, higher frequency IDs correspond to lower angular frequencies. This mapping allows us to analyze the contribution of each frequency to the model's binding behavior.

We first evaluate the metrics separately for each frequency. Figure 6 shows the location of selected attention heads in the positional–symbolic plane, together with their detailed frequency profiles. Since the model uses a head dimension of 256, this corresponds to 128 RoPE frequencies. By convention, higher frequency IDs correspond to lower angular frequencies, while lower frequency IDs correspond to higher angular frequencies.

We then aggregate these measurements by summing logits across frequencies, yielding a global positional–symbolic profile. The results, presented in Figure 1, 7, provide an overall view across all layers and heads of the model.

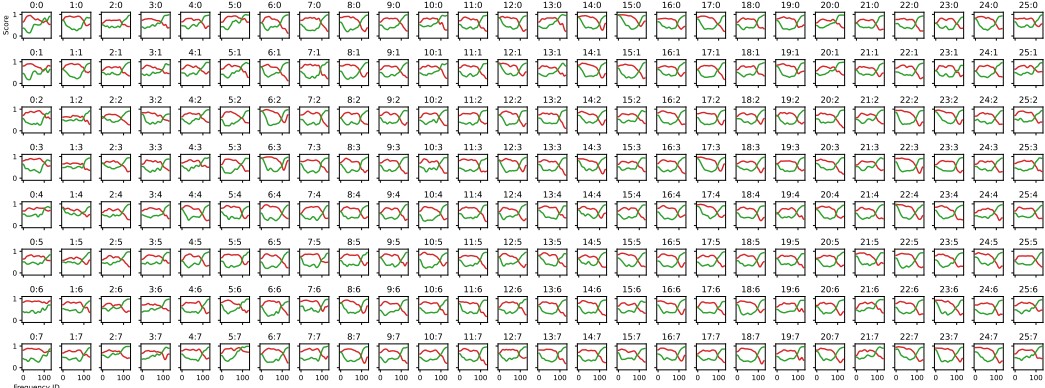

Figure 5: **Positional–symbolic profiles of all attention heads in `google/gemma-2-2b-it`.** Each subplot corresponds to an attention head, 26 layers, 8 heads per layer.

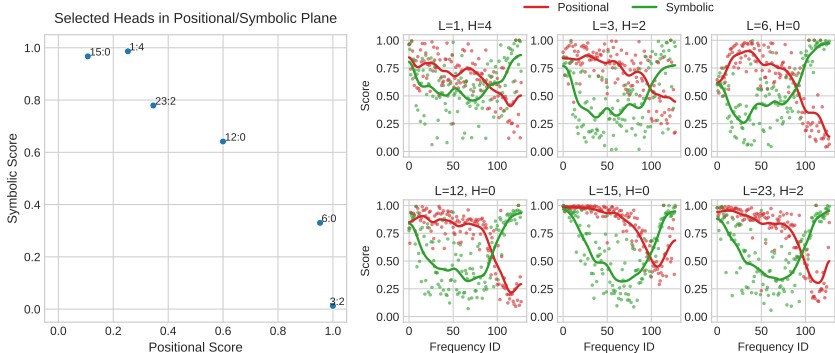

Figure 6: **Positional–Symbolic Plane across Frequencies.** Visualization of six attention heads from the `google/gemma-2-2b-it` model. (Left) Each point represents a head in the positional–symbolic plane, labeled by its layer and head index. (Right) For the same heads, we plot their positional and symbolic scores as a function of RoPE frequencies.

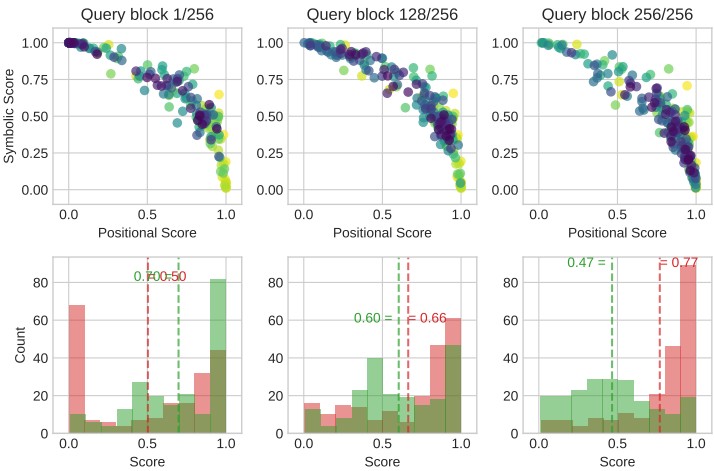

Figure 7: **Heads when the block moves (1 to 9).** (Above) Heads in the positional–symbolic plane for each block position. (Bellow) Histograms showing how positional and symbolic scores change as the relevant block moves.

# C   ADDITIONAL ILLUSTRATION AND PROOFS OF SECTION 4

## C.1   ILLUSTRATION OF CANONICAL TASKS

See Figure 8 for an illustration of how the canonical tasks presented in Section 4 should work.

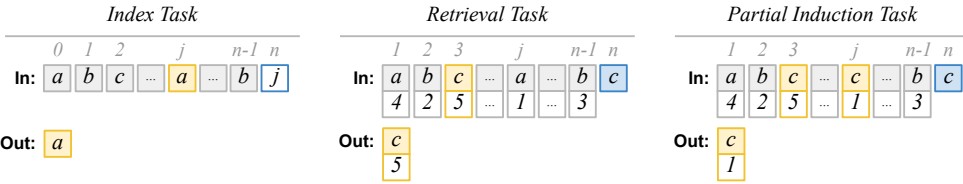

Figure 8: Illustration of the canonical tasks.

## C.2 PROOF OF THEOREM 2

First, we will prove that whenever a head $H = (L, \text{VAL}, F)$ acts symbolically on input $\bar{x}$, the output of the model $M_{\text{EMB}}$ is invariant over permutations of the input.

**Lemma 1.** *For a fixed vocabulary $\Sigma = \{s_1, s_2, ..., s_{|\Sigma|}\}$ and input $s^* = (s_{i_1}, ..., s_{i_n})$ such that $H$ acts symbolically on $\text{EMB}(s^*)$, then*

$$M_{\text{EMB}}(s^*) = M_{\text{EMB}}(\pi(s^*)) \qquad \forall \pi \in S^{n-1}$$

*Proof.* Set $\text{EMB}(s^*) = (x_1, ..., x_n)$ and fix a position $k \leq n$. Now, we have that the attention weight at position $k$ with input $s^*$ is

$$w_{nk}(s^*) = \frac{\exp(L(x_n, n, x_k, k))}{\displaystyle\sum_{l \leq n} \exp(L(x_n, n, x_l, l))} = \frac{\exp(L(x_n, n, x_k, \pi(k)))}{\displaystyle\sum_{l \leq n} \exp(L(x_n, n, x_{\pi(l)}, \pi(l)))} = w_{n\pi(k)}(\pi(s^*)),$$

because $H$ acts symbolically on $s^*$. Now, if we look at the attention vector for the last position with input $s^*$ we get

$$a_n(s^*) = \frac{\displaystyle\sum_{j \leq n} w_{nj}(s^*)\text{VAL}(x_j)}{\sum_{j \leq n} w_{nj}(s^*)} = \frac{\displaystyle\sum_{j \leq n} w_{n\pi(j)}(s^*)\text{VAL}(x_{\pi(j)})}{\sum_{j \leq n} w_{n\pi(j)}(s^*)} \tag{2}$$

$$= \frac{\displaystyle\sum_{j \leq n} w_{n\pi(j)}(\pi(s^*))\text{VAL}(x_{\pi(j)})}{\sum_{j \leq n} w_{n\pi(j)}(\pi(s^*))} \tag{3}$$

$$= a_n(\pi(s^*)). \tag{4}$$

Which implies that $M_{\text{EMB}}$ has the same output for $s^*$ and $\pi(s^*)$. $\qquad\square$

Consider an arbitrary vocabulary $\Sigma$ and $s \in \Sigma^*$ such that $\bar{x} = \text{EMB}(s) = (x_1, x_2, ..., x_n)$ is the input where $H$ acts symbolically. Since $\bar{x}$ is not constant, there exists a permutation $\pi \in S^{n-1}$ such that $f_{\text{POS}}(\pi(s)) \neq f_{\text{POS}}(s)$ (we just change what is in the position $j$ being asked). Then, from Lemma 1 it follows that $M_{\text{EMB}}(s) \neq f_{\text{POS}}(s)$ or $M_{\text{EMB}}(\pi(s)) \neq f_{\text{POS}}(\pi(s))$.

## C.3 PROOF OF THEOREM 3

The goal is to design an attention head $H_{\text{POS}}$ such that the model $M_{\text{ONEHOT}}$ using $H_{\text{POS}}$ returns the symbol $\sigma_j$ at position $j$ with input $s_{\text{POS}}$, i.e. it performs an indexing operation.

We fix a sequence length $n > 0$, and define the desired inner product condition:

$$x_i K^\top R_\theta^{n-i} Q x_n(j) = v^\top R_\theta^{n-i} q_n(j) = \cos(\theta(j - i))$$

for $i = 0, ..., n$.

In order to achieve this, first we fix a vector $v = (1/2, 1/2)^\top$ and the rotation angle $\theta = \pi/n$ (so maximum is reached just for $i = j$). We define the query vector as:

$$q_n(j) = R_\theta^{j-n} v$$

Therefore, we get a query matrix $Q_{\text{POS}}$ such that:

$$Q_{\text{POS}} e_j = q_n(j)$$

where $e_j$ is the one-hot encoding of the symbol $j$. For our solution, we use the ONEHOT embedding function. On the other hand, we define a constant key matrix $K_{\text{POS}}$ such that for any position $i$, the key embedding of the symbol at that position is:

$$K_{\text{POS}}\text{ONEHOT}(\sigma_i) = v.$$

With these choices, the result is a one-layer transformer $M_{\text{ONEHOT}}$ with attention head $H_{\text{POS}} = (K_{\text{POS}}, Q_{\text{POS}}, Id, \pi_1)$ that always maximizes its attention at the right position. Therefore, $M_{\text{ONEHOT}}$ perfectly copies the symbol at the position asked by an input $s_{\text{POS}}$, at least when $s_{\text{POS}}$ has length $n$. Now, by Theorem 3 we know that $H_{\text{POS}}$ can not be a symbolic head for any input. We formally state its positional nature in the following lemma.

**Lemma 2.** *For every input $s_{\text{POS}}$, $H_{\text{POS}}$ is a positional head on* $\text{ONEHOT}(s_{\text{POS}})$.

*Proof.* We just apply the definition of a positional head on $y, z \in \{x_1, \dots, x_{n-1}\}, i \in [n-1]$ then

$$L(x_n(j), n, y, i) = \cos\left(\theta(j-i)\right) = L(x_n(j), n, z, i).$$

$\square$

**Lemma 3.** *For the head $H_{\text{POS}}$ and any $s_{\text{POS}}$ with $|\text{ONEHOT}(s_{\text{POS}})| = n$ then*

$$M_{\text{ONEHOT}}(s_{\text{POS}}) = f_{\text{POS}}(s_{\text{POS}}).$$

*Proof.* First, we see that logits are maximized at position $j$.

$$\arg\max_{i \in [n-1]} L(x_n(j), n, x_i, i) = \arg\max_{i \in [n-1]} \cos(\theta(j-i)) = j.$$

.

Therefore, the biggest attention weight is $w_n^j$, i.e

$$\max\{w_n^i\}_{i=1}^n = w_n^j.$$

Now, given that the value function $\text{VAL} = id$, then we have that

$$a_n = \sum_{i=1}^n w_n^i x_i,$$

which implies that $\arg\max_{\sigma_i \in \Sigma} \text{SOFTMAX}(a_n)_i = \text{ONEHOT}^{-1}(x_j) = \sigma_j$ concluding the proof. $\square$

### C.4 PROOF OF THEOREM 4

Suppose there exist an input $s_{\text{SYM}} = (s_1, \dots, s_n)$, an attention head $H = (Q, K, U, id, \pi_1)$ that acts positionally on $\bar{x} = \text{ONEHOT}(s_{\text{SYM}}(H))$ and $M_{\text{ONEHOT}} = f_{\text{SYM}}$. Now, by defining the function

$$D(x_n, n, x_j, j) = \frac{\exp(L(x_n, n, x_j, j))}{\sum_{k \leq n} \exp(L(x_n, n, x_k, k))},$$

we can express the attention vector $a_n(s_{\text{SYM}}(H))$ as

$$a_n(s_{\text{SYM}}(H)) = \sum_{j \leq n} D(x_n, n, x_j, j)x_j.$$

We also have that

$$M_{\text{ONEHOT}}(s_{\text{SYM}}) = \arg\max_{\sigma_i \in \Sigma} \text{SOFTMAX}(\pi_1(a_n, x_n))_i \tag{5}$$

$$= \text{ONEHOT}^{-1}\left(\arg\max_{x_j} D(x_n, n, x_j, j)\right) \tag{6}$$

$$= s_{i^*} \tag{7}$$

by exploiting the fact that $\{x_j\}_{j=1}^n$ is an orthonormal basis and $\pi_1(a_n, x_n) = a_n$. In this case, $i^*$ is the position of the input $s_{\text{SYM}}(H)$ where the answer appears. Since $M_{\text{ONEHOT}}$ solves $f_{\text{SYM}}$, there is a unique vector $x_{i^*}$ that maximizes the output distribution. Therefore, there exists a position $k < n$ with $k \neq i^*$ such that

$$D(x_n, n, x_k, k) < D(x_n, n, x_{i^*}, i^*).$$

Now, applying the positional behavior of $H$, we get that

$$D(x_n, n, x_{i^*}, i^*) = D(x_n, n, x_k, i^*) \quad \text{and} \quad D(x_n, n, x_k, k) = D(x_n, n, x_{i^*}, k).$$

Finally, consider a permutation $\pi_{i^* \to k} \in S^{n-1}$ that just interchange $s_{i^*}$ with $s_k$. We have that

$$a_n(\pi_{i^* \to k}(s_{\text{SYM}}(H))) = \sum_{j \neq i^*} D(x_n, n, x_j, j)x_j + D(x_n, n, x_k, i^*)x_k,$$

which implies that

$$M_{\text{ONEHOT}}(\pi_{i^* \to k}(s_{\text{SYM}})) = \arg\max_{\sigma_i \in \Sigma} \text{SOFTMAX}(\pi_1(a_n, x_n))_i \qquad (8)$$

$$= \text{ONEHOT}^{-1}(\arg\max_{x_j} D(x_n, n, x_j, j)) \qquad (9)$$

$$= s_k \qquad (10)$$

And therefore,

$$M_{\text{ONEHOT}}(s_{\text{SYM}}(H)) \neq M_{\text{ONEHOT}}(\pi_{i^* \to k}(s_{\text{SYM}}(H))),$$

but $f_{\text{SYM}}(s_{\text{SYM}}(H)) = f_{\text{SYM}}(\pi_{i^* \to k}(s_{\text{SYM}}(H)))$ which is a contradiction.

### C.5 PROOF OF THEOREM 5

Fix a sequence length $n > 0$ and a vocabulary $\Sigma = \{\sigma_1\#, \sigma_1\#1, \sigma_1\#2, ..., \sigma_m\#k\}$ with $m, k > 0$. We want a matrix $K_{\text{SYM}}$ such that for each symbol $\sigma_i$ there exist a vector

$$v_{\sigma_i} = K_{\text{SYM}}\text{ONEHOT}(\sigma_i\#\alpha)$$

for any $\alpha \in \{1, ..., k\}$ and $K_{\text{SYM}}\text{ONEHOT}(\sigma_i\#) = 0$ for any $\sigma_i$.

On the other hand, we want a matrix $Q_{\text{SYM}}$ such that

$$Q_{\text{SYM}}x_n(\sigma_j\#) = q_n(\sigma_j\#) = R_\theta^{l-n}v_{\sigma_j}.$$

In essence, we want that vectors $v_{\sigma_i} \in \mathbb{R}^2$ codify all symbols $s \in \Sigma$ such that $\sigma_i$ appears in $s$. In general, this can be done for $\theta < \frac{2\pi}{m}$ in order to avoid overlapping in the vectors $v_{\sigma_i}$ after the rotation with $R_\theta$. For example, we can fix the vector $v_{\sigma_1} = (1, 0)^\top$ and an angle $\beta = \frac{2\pi}{m}$. Then, for $i \in [2, ..., m]$ we have that $v_{\sigma_i} = R_\beta^i v_{\sigma_1}$.

Having such matrices for a fixed $\theta$, we get a logits function defined as

$$L(\text{ONEHOT}(\sigma_j\#), n, \text{ONEHOT}(\sigma_i\#\alpha), t) = x_t^T K_{\text{SYM}}^\top R_\theta^{n-t} Q_{\text{SYM}} x_n \qquad (11)$$

$$= v_{\sigma_i}^\top R_\theta^{n-t} q_n(\sigma_j\#) \qquad (12)$$

$$= v_{\sigma_i}^\top R_\theta^{l-t} v_{\sigma_j} \qquad (13)$$

for $t = 0, \ldots, n-1$.

We refer to the attention head using this logits function as $H_{\text{SYM}}^\theta$. Notice that when $\theta = 0$, $H_{\text{SYM}}^\theta$ uses the *NoPE* (No Positional Encoding) since the operator $R_\theta$ becomes the identity. In consequence, $H_{\text{SYM}}^0$ is purely symbolic for any input.

In the case that $\theta > 0$, $R_\theta$ introduces a rotation that perturbs the query vectors. To maintain symmetry of the inner product structure, the vectors must be centered in the middle of the context. This requirement naturally leads to the choice $l = \frac{n-1}{2}$. However, in this case the $H_{\text{SYM}}^\theta$ does not act symbolically given that logits depends on the positions.

To formalize our construction for $\theta = 0$, we prove the following lemma.

**Lemma 4.** *For the head $H_{\text{SYM}}^0$ and any $s_{\text{SYM}}$ with $|\text{ONEHOT}(s_{\text{SYM}})| = n$ then*

$$M_{\text{ONEHOT}}(s_{\text{SYM}}) = f_{\text{SYM}}(s_{\text{SYM}})$$

*Proof.* Using the fact that

$$\arg\max_{i \in [n-1]} L(x_n(\sigma_j), n, x_i, i) = \arg\max_{i \in [n-1]} v_{\sigma_i}^\top v_{\sigma_j} = j.$$

and the argument follows from the same idea of the proof of Lemma 3. □

Finally, by Theorem 4 we know that $H^0_{\text{SYM}}$ can not be a positional head. Now we proceed to show that it is indeed a symbolic one.

**Lemma 5.** *For every input $s_{\text{SYM}}$, the head $H^0_{\text{SYM}}$ is symbolic on* $\text{ONEHOT}(s_{\text{SYM}})$.

*Proof.* For $i, j \in [m], y = \text{ONEHOT}(\sigma_i \# \alpha)$ and $, k_1, k_2 \in [n-1]$

$$L(x_n(\sigma_j\#), n, y, k_1) = v_{\sigma_i}^\top v_{\sigma_j} = L(x_n(\sigma_j\#), n, y, k_2).$$

$\square$

## C.6   A COUNTER EXAMPLE WITH MLP LAYER

**Counter Example.**   Fix the sequence length to be $n = 3$ and consider a vocabulary consisting of symbols $\{a, b\}$ and integers $\{0, 1\}$. We take as input a sequence $s_{\text{SYM}}$ with associated embedded soft-vectors $\overline{x}$,

$$s_{\text{SYM}} = (\sigma_1\#i_1,\ \sigma_2\#i_2,\ \sigma\#), \quad \overline{x} = \text{ONEHOT}(s_{\text{SYM}}) = (y_1, y_2, x).$$

**Attention Head Construction.**   Consider a head $H$ with logit function $L$ satisfying:

$$L(x, 3, y_1, 1) = \cos(2\pi) = 1, \quad L(x, 3, y_2, 2) = \cos(\pi) = -1, \quad L(x, 3, x, 3) = 0.$$

This can be achieved by choosing RoPE with $\theta = \pi$, and matrices $Q, K$ such that

$$Qx = Ky_j = (1, 0, 0, 0, 0, 0) \text{ (for positions } j = 1, 2), \quad Kx = (0, 0, 0, 0, 0, 0) \text{ (at the last token)}.$$

With $\text{VAL} = id$, the attention vector at the last position is

$$a_3 = \big(\gamma(a\#1, \overline{x}), \gamma(a\#2, \overline{x}), \gamma(b\#1, \overline{x}), \gamma(b\#2, \overline{x}), \gamma(a\#, \overline{x}), \gamma(b\#, \overline{x})\big),$$

where $\gamma$ is defined as follows

$$\gamma(\sigma_j\#, \overline{x}) = \lambda_3 \cdot \mathbb{I}(\sigma = \sigma_j), \quad \gamma(\sigma_j\#i_j, \overline{x}) = \begin{cases} \frac{e+e^{-1}}{\Sigma} & \text{if } \sigma_j\#i_j \text{ occurs twice,} \\ 0 & \text{if } \sigma_j\#i_j \text{ never occurs,} \\ \lambda_j & \text{if } \sigma_j\#i_j \text{ occurs exactly once.} \end{cases}$$

where $\lambda_1 = \frac{e}{\Sigma}$, $\lambda_2 = \frac{e^{-1}}{\Sigma}$ and $\lambda_3 = \frac{1}{\Sigma}$, with $\Sigma = e + e^{-1} + 1$.

**Final Output Function.**   Define the function $F(a_3) \in \mathbb{R}^6$ by

$$F(a_3)_{\sigma_j\#i_j} = \text{AND}(\text{EQ}(\gamma(\sigma_j\#, \overline{x}), \lambda_3), \text{OR}(\text{EQ}(\gamma(\sigma_j\#i_j, \overline{x}), \lambda_1), \text{EQ}(\gamma(\sigma_j\#i_j, \overline{x}), \lambda_2)))$$

and $F(a_3)_{\sigma_j\#} = 0$. The logical operations AND, OR and EQ can be implemented by an MLP with ReLU activations and fixed precision with sufficient depth (see Lemma A.1 in Yao et al. (2021)).

## C.7   PROOF OF COROLLARY 1

By Lemma 1 we know that if $H$ acts symbolically on an input $s_{\text{MIX}}(H) = (\sigma_0\#i_0, \sigma_1\#i_1, \ldots, \sigma_{n-1}\#i_{n-1}, \sigma_j\#)$, then for every permutation of $s_{\text{MIX}}(H)$ the answer of the one-head decoder-only transformer model $M_{\text{ONEHOT}}$ using $H$ is invariant. However, the answer of $f_{\text{MIX}}$ must change if we permute both appearances of $\sigma_j$ in $s_{\text{MIX}}(H)$ which means $M_{\text{ONEHOT}}$ can not solve $f_{\text{MIX}}$.

On the other hand, by using the same argument as the proof C.4, we know that if $H$ acts positionally on $s_{\text{MIX}}$, then we can force its model's output to change for permutations of $s_{\text{MIX}}$ where $f_{\text{MIX}}$ is invariant.

C.8    PROOF OF THEOREM 6

In this case we mix the ideas we used to define heads $H_{\text{POS}}$ and $H_{\text{SYM}}^\theta$, by using a RoPE with 2 frequencies. First, fix the vocabulary $\Sigma = \{\sigma_1\#, \sigma_1\#1, \sigma_1\#2, ..., \sigma_m\#k\}$ with $m, k > 0$. In the same way as we did for $H_{\text{SYM}}^\theta$ we can define vectors $v_{\sigma_i} \in \mathbb{R}^2$ that codify the presence of $\sigma_i$ in a symbol of $\Sigma$. On the other hand, as we did for $H_{\text{POS}}$, we set a constant vector $v = (1/2, 1/2)^\top$. Now, we define a matrix $K \in \mathbb{R}^{4 \times |\Sigma|}$ such that

$$K_{\text{MIX}}\text{ONEHOT}(\sigma_1\#\alpha) = [v_{\sigma_i}, v],$$

for any $\alpha \in \{1, ..., k\}$, while $\text{ONEHOT}(\sigma_i\#)K_{\text{MIX}} = 0$ for any $i \in \{1, ..., m\}$.

For the query matrix $Q_{\text{MIX}}$, we define a query matrix such that

$$Q_{\text{MIX}}\text{ONEHOT}(\sigma_j\#) = q_n(\sigma_j\#) = [R_\theta^{l-n}v_{\sigma_j}, v].$$

Finally, for angles $\theta_1, \theta_2$, we get a logits function

$$
\begin{aligned}
\text{ONEHOT}(\sigma_i\#\alpha)K_{\text{MIX}}^\top R_\Theta^{n-i} Q_{\text{MIX}}\text{ONEHOT}(\sigma_j\#) &= v_{\sigma_i}^\top R_{\theta_1}^{n-i}q_n(\sigma_j) + v^\top R_{\theta_2}^{n-i}v && (14) \\
&= v_{\sigma_i}^\top R_{\theta_1}^{l-i}v_{\sigma_j} + \cos\left(\theta_2(n-i)\right) && (15)
\end{aligned}
$$

If we set $\theta_1 = 0$ (NoPE), then our logits function becomes

$$L(\text{ONEHOT}(\sigma_j\#), n, \text{ONEHOT}(\sigma_i\#\alpha), k) = v_{\sigma_i}^\top v_{\sigma_j} + \cos\left(\theta_2(n-k)\right),$$

with $0 < \theta_2 n < 2\pi/(n|\Sigma|)$ properly chosen to define our solution head $H_{\text{MIX}}^{0,\theta_2}$.

**Lemma 6.** *For $H_{\text{MIX}}^{0,\theta_2}$ and any $s_{\text{MIX}}$ with $|\text{ONEHOT}(s_{\text{MIX}})| = n$ then $M_{\text{ONEHOT}}(s_{\text{MIX}}) = f_{\text{MIX}}(s_{\text{MIX}})$.*

*Proof.* Notice that:
$$\theta_2 < 2\pi/(n|\Sigma|) \Rightarrow \cos(2\pi/|\Sigma|) < \cos\left(\theta_2 n\right)$$
and $v_{\sigma_i}^\top v_{\sigma_j} < \cos(2\pi/|\Sigma|)$ for $\sigma_i \neq \sigma_j$. Then

$$
\begin{aligned}
v_{\sigma_i}^\top v_{\sigma_j} + \cos\left(\theta_2(n-k)\right) &\leq \cos(2\pi/|\Sigma|) + \cos\left(\theta_2(n-k)\right) \\
&< \cos\left(\theta_2 n\right) + \cos\left(\theta_2(n-k)\right) \\
&\leq \cos\left(\theta_2 n\right) + 1 \\
&\leq \cos\left(\theta_2(n-j_p)\right) + 1 \\
&= \cos\left(\theta_2(n-j_p)\right) + v_{\sigma_{j_p}}^\top v_{\sigma_j}
\end{aligned}
$$

for all $k \in [n-1] \setminus \{j_1, j_2\}$ with $p = 1, 2$, where $v_{\sigma_{j_1}} = v_{\sigma_{j_2}} = v_{\sigma_j}$.

Using this and $j_1 < j_2$, we can conclude:

$$
\begin{aligned}
\arg\max_{i \in [n-1]} L(x_n(j), n, x_i, i) &= \arg\max_{i \in [n-1]} \left(v_{\sigma_i}^\top v_{\sigma_j} + \cos\left(\theta_2(n-i)\right)\right) \\
&= \arg\max_{i \in \{j_1, j_2\}} \left(v_{\sigma_i}^\top v_{\sigma_j} + \cos\left(\theta_2(n-i)\right)\right) \\
&= \arg\max_{i \in \{j_1, j_2\}} \cos\left(\theta_2(n-i)\right) \\
&= j_2.
\end{aligned}
$$

And the result follows from the fact that the most attended symbol of the input is the output for this architecture as seen in the proof of Lemma 3. $\square$

# D  APPENDIX FOR SECTION 5

## D.1  SPECIALIZED HEADS FOR INDEX, INFORMATION RETRIEVAL AND PARTIAL INDUCTION TASKS

To further disentangle the role of positional and symbolic information, we train from scratch a family of minimal transformer models, each consisting of a single layer, a single attention head, and a single RoPE frequency. Each model is trained independently with a different frequency choice, and performance is evaluated on two tasks: an *index task* and an *information retrieval task* (see Section 4 for precise definitions).

All experiments are carried out on sequences of length $n = 32 + 1$, where 32 tokens form the context and the additional token is the query. Both tasks are defined over mixed symbol–integer sequences. In the *information retrieval task*, the model must retrieve the integer associated with the query symbol, while in the *index task*, the model must instead return the symbol associated with the query integer (which directly encodes a position). We fix a vocabulary of 16 symbols and 32 possible integers.

We sweep over a range of base angles for the RoPE:

$$[0.0,\ 0.01,\ 0.1,\ 0.25,\ 0.4,\ 0.5,\ 0.8,\ 1.0,\ 1.5,\ 2.0].$$

Here, $0.0$ corresponds to a NoPE model (no rotary position encoding), while $2.0$ corresponds to a RoPE configuration that completes one full angular lap over the $n = 33$ tokens of the sequence. More generally, the effective frequency is given by

$$\theta = \frac{\pi}{n} \cdot \text{base\_angle},$$

and the number of laps is defined as $n\theta/2\pi$.

Training is performed for 100 epochs with batch size 64, using $40{,}960$ training samples and $20{,}480$ validation samples. Figure 2 reports the resulting accuracies as a function of frequency. The results show a clear tension: index task and information retrieval achieve their best performance at different frequency ranges, indicating that the model's inductive bias over RoPE frequencies favors one type of task at the expense of the other.

## D.2  PROOF OF THEOREM 7: U-SHAPE AND INVERTED U-SHAPE.

**U-shape accuracy.**  Let us start with the statement $w_{\max}$ is $U$-shaped for $H_{\text{POS}}$. As established in the proof of Theorem 3, the logits of $H_{\text{POS}}$ are of the form:

$$L(x_n, n, x_i, i) = \cos(\theta(j - i)),$$

where $\theta < \frac{2\pi}{n}$ and $j$ is the location of the answer. Thus, we obtain the following expression for $w_{\max}(j)$:

$$w_{\max}(j) = \frac{\exp(1)}{\exp(\cos(\theta(j-1)) + \ldots + \exp(\cos(\theta(j-n)))}.$$

What happens when we go from $w_{\max}(j)$ to $w_{\max}(j + 1)$? We add $\exp(\cos(\theta j))$ and delete $\exp(\cos(\theta(j-n)))$ from the denumerator. That is, if $\cos(\theta j) > \cos(\theta(j-n))$, the value of $w_{\max}(j)$ decreases, and otherwise, it stays the same or increases.

Since $\theta < \frac{2\pi}{n}$, we have $\cos(\theta j) > \cos(\theta(j - n))$ if and only if $|j| < |j - n|$. Hence, starting from $j = 1$ when $|j|$ is much smaller than $|j - n|$, the function $w_{\max}$ is decreasing, having a local maximum at $j = 1$. Weh $j$ is about $n/2$, the absolute values $|j|$ and $|j - n|$ become equal and then the first one becomes larger, and afterwards the function $w_{\max}$ starts increasing, reaching another local maximum at $j = n - 1$ in the form of a $U$-shape, as required.

**Inverted U-shape accuracy.**  We consider a simple RoPE head with one angle, equal to $\pi/n$, as the one used for the empirical results depicted in Figure 3 where the inverted U-shape is observed in the retrieving (symbolic) task. We will rigorously prove this is the case in a simplified scenario. The simplified task will be the following: we have two possible tokens $\tau_0, \tau_1$ and we consider prompts $x_1 \ldots x_n$ consisting of $n$ copies of $\tau_0$, except at two positions where it has $\tau_1$. These two positions

are the query ($j = n$) and some $\ell \le m = n - 1$. The task for the attention head is to detect the requested token in position $\ell$, which means that we expect a distribution of attention wights $\alpha_j$ ($j = 1, ..., m$) with small values except for a peak at $j = \ell$. This must hold for every prompt of this form.

As we have empirically demonstrated, RoPE cannot be expected to behave symbolically for this choice of angle (it is too big) so we accept a weak solution of the symbolic task: the logit sequence $\lambda_1, ..., \lambda_m$ is negative, except for a positive value when we reach the desired token $\tau_1$. We also want that the logits are bounded away from 0 (positive or negative) as much as possible. Since the query position is fixed, if $q = Q x_n \in \mathbb{R}^2$ is the query vector, then we can write the logit function as

$$L(x_n, n, x, j) = L(x, j) = xKU^j q$$

($U$ is the RoPE matrix for angle $\pi/n$). For any embedding vector $x \in \mathbb{R}^d$ let $x' = xK \in \mathbb{R}^2$. Let $z_i$ be the embedding vector of $\tau_i$ for $i = 0, 1$. We make the simplifying assumption that $q$, $z_0'$ and $z_1'$ have norm 1 —the discussion can be adapted to a more general case, but we choose to keep the computations simple while still exhibiting the main features of the inverted U-shape phenomenon.

Let $\gamma(x)$ be the angle between $q$ and the vector $x' = xK$ for $x \in \mathbb{R}^d$. Then the formula for the RoPE logit and the assumption $\|q\| = 1$ gives

$$L(x, j) = \|x'\| \cos(\pi j/n - \gamma(x)).$$

In particular, our assumption that this logit function gives an approximate (in the sense of sign) solution to the symbolic task described above, with logits bounded away from 0 as much as possible, gives

$$\gamma(z_0) = -\pi/2, \quad \gamma(z_1) = \pi/2$$

so that

$$L(z_i, j) = \begin{cases} \cos(\pi j/n + \pi/2) & \text{if } i = 0 \\ \cos(\pi j/n - \pi/2) & \text{if } i = 1 \end{cases}$$

(here we used the simplifying assumption $\|z_i'\| = 1$). We note the symmetry

$$L(z_1, j) = -L(z_0, j).$$

Let

$$\nu_i(j) = \exp(L(z_i, j))$$

and note that $\nu_0(\ell)\nu_1(\ell) = 1$. Define

$$S = \sum_{j=1}^{n-1} \nu_0(j).$$

Recall that the position of the sought token is $\ell$. Then the attention weight at $j = \ell$ is maximal (due to the sign of the logits –this is the only positive logit for the given prompt) and it is

$$w_{\max}(\ell) = \frac{\nu_1(\ell)}{S + \nu_1(\ell) - \nu_0(\ell)}.$$

This is the only peak of the attention weights for the prompt with sought token at position $\ell$.

Finally, we can give a rigorous proof of the inverted U-shape phenomenon in this setting.

**Theorem 8** (Inverted U-shape). *Assume $n \ge 5$. As a function of $\ell$, the sequence $w_{\max}(\ell)$ is increasing for $0 < \ell < n/2$ and decreasing for $n/2 < \ell < n$.*

*Proof.* We can use the same formula defining $w_{\max}(\ell)$ on a real variable $t$ instead of the discrete variable $\ell$, and then restrict back to integers $\ell$.

Let $f(t) = 1/w_{\max}(t)$. It suffices to prove that $f'(t)$ is negative for $0 < t < n/2$ and positive for $n/2 < t < n$.

From the equation $\nu_0(\ell)\nu_1(\ell) = 1$ we have

$$f(t) = 1 + S\nu_0(t) - \nu_0(t)^2.$$

Then

$$f'(t) = S\nu_0'(t) - 2\nu_0'(t)\nu_0(t) = \nu_0'(t)\left(S - 2\nu_0(t)\right).$$

If $n \geq 5$ and $t = \ell$ is an integer, the parenthesis is strictly positive because we are substracting 2 different sumands from $S$ (here we use $n$ is odd). On the other hand, we explicitly compute $\nu_0'(t)$:

$$\nu_0'(t) = -\frac{\pi}{n}\nu_0(t)\sin(\pi t/n + \pi/2).$$

As $\nu_0(t) > 0$ always, the sign of the previous expression is negative for $\pi t/n \in (0, \pi/2)$ and positive for $\pi t/n \in (\pi/2, \pi)$. The result follows. $\qquad\square$

# E  ADDITIONAL RESULTS FOR REALISTIC MODELS

## E.1  LLAMA-3.2-1B-INSTRUCT

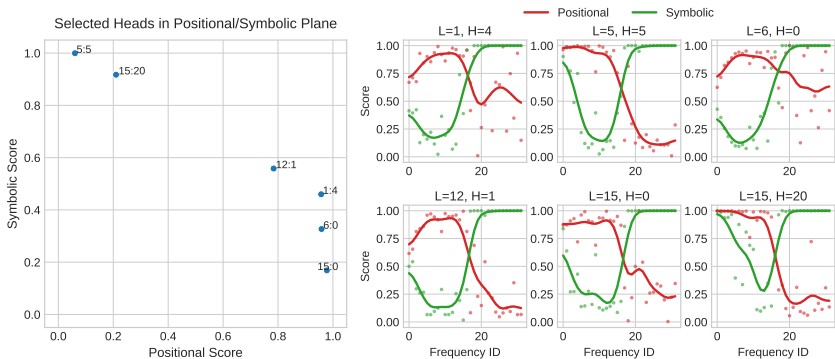

(a) **Positional–Symbolic Plane across Frequencies.**  Visualization of six attention heads from the `meta-llama/Llama-3.2-1B-Instruct` model.  (Left) Each point represents a head in the positional–symbolic plane, labeled by its layer and head index. (Right) For the same heads, we plot their positional and symbolic scores as a function of RoPE frequencies.

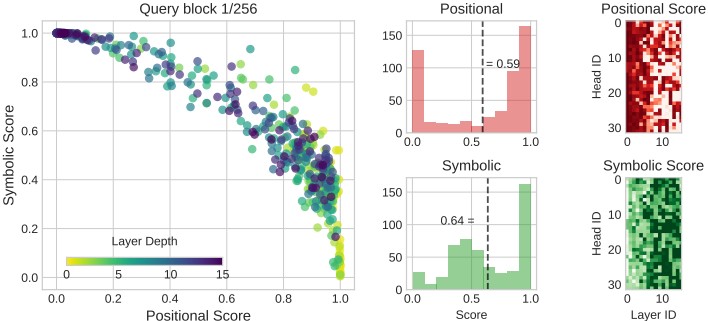

(b) **All heads, fixed first block.**  (Left) Each head in the positional–symbolic plane. (Middle) Histogram showing predominance of symbolic over positional behavior. (Right) Heatmaps of positional and symbolic scores for each head across layers.

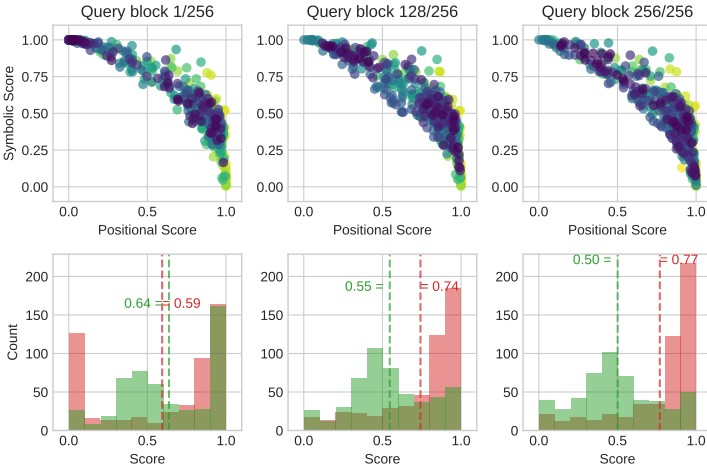

(c) **Heads when the block moves (1 to 9).**  (Above) Heads in the positional–symbolic plane for each block position. (Bellow) Histograms showing how positional and symbolic scores change as the relevant block moves.

Figure 9: **Positional–Symbolic Plane across Heads.**  16 Layers, 32 Heads per layer.  Since the model uses a head dimension of 64, this corresponds to 32 RoPE frequencies.

### E.1.1 LLAMA-3.2-3B-INSTRUCT

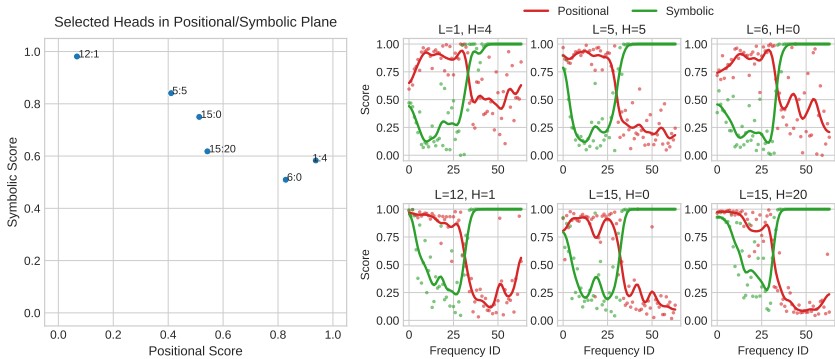

(a) **Positional–Symbolic Plane across Frequencies.** Visualization of six attention heads from the `meta-llama/Llama-3.2-3B-Instruct` model. (Left) Each point represents a head in the positional–symbolic plane, labeled by its layer and head index. (Right) For the same heads, we plot their positional and symbolic scores as a function of RoPE frequencies.

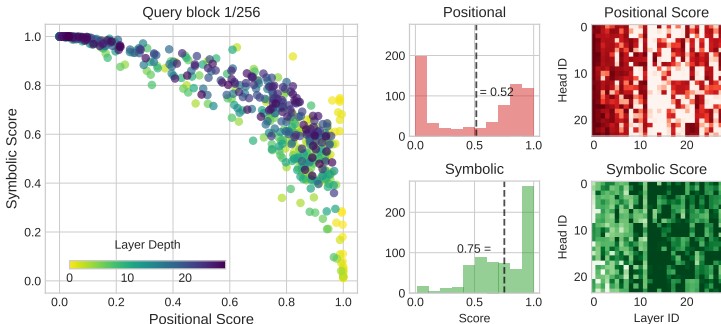

(b) **All heads, fixed first block.** (Left) Each head in the positional–symbolic plane. (Middle) Histogram showing predominance of symbolic over positional behavior. (Right) Heatmaps of positional and symbolic scores for each head across layers.

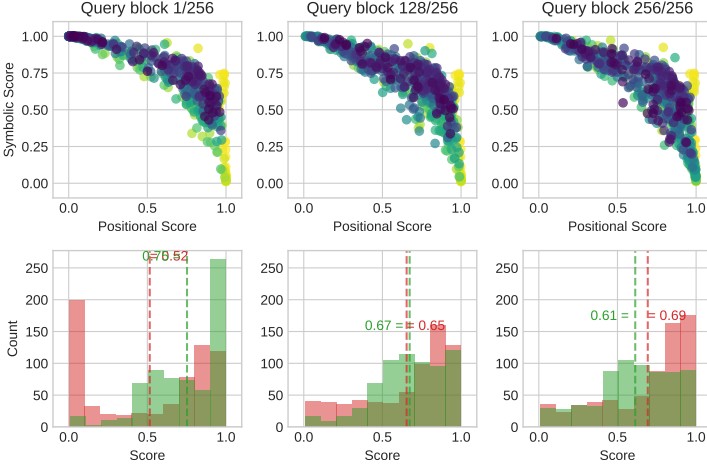

(c) **Heads when the block moves (1 to 9).** (Above) Heads in the positional–symbolic plane for each block position. (Bellow) Histograms showing how positional and symbolic scores change as the relevant block moves.

Figure 10: **Positional–Symbolic Plane across Heads.** 28 Layers, 24 Heads per layer. Since the model uses a head dimension of 128, this corresponds to 64 RoPE frequencies.

## E.2 QWEN2-1.5B-INSTRUCT

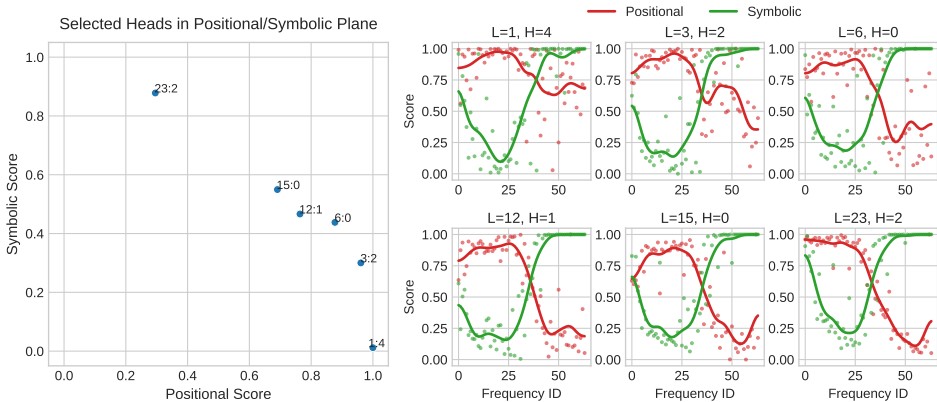

(a) **Positional–Symbolic Plane across Frequencies.** Visualization of six attention heads from the `qwen/Qwen2-1.5B-Instruct` model. (Left) Each point represents a head in the positional–symbolic plane, labeled by its layer and head index. (Right) For the same heads, we plot their positional and symbolic scores as a function of RoPE frequencies.

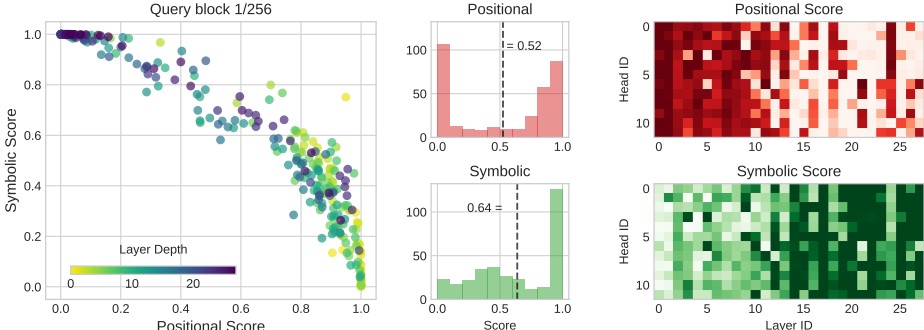

(b) **All heads, fixed first block.** (Left) Each head in the positional–symbolic plane. (Middle) Histogram showing predominance of symbolic over positional behavior. (Right) Heatmaps of positional and symbolic scores for each head across layers.

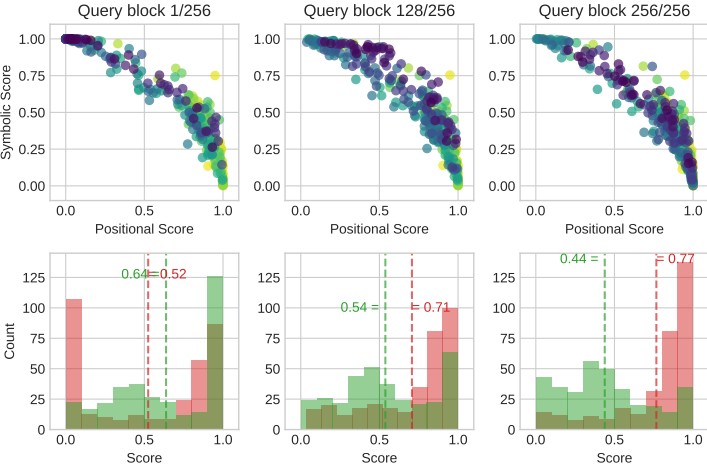

(c) **Heads when the block moves (1 to 9).** (Above) Heads in the positional–symbolic plane for each block position. (Bellow) Histograms showing how positional and symbolic scores change as the relevant block moves.

Figure 11: **Positional–Symbolic Plane across Heads.** 28 Layers, 12 Heads per layer. Since the model uses a head dimension of 128, this corresponds to 64 RoPE frequencies.

### E.3 QWEN2-0.5B-INSTRUCT

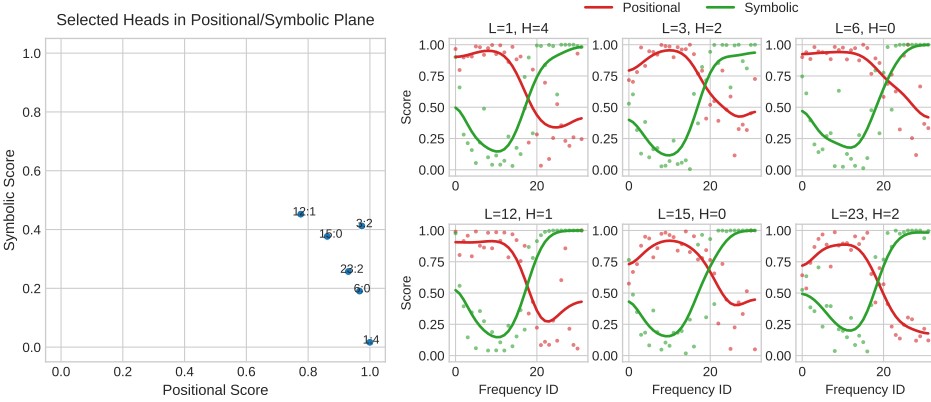

(a) **Positional–Symbolic Plane across Frequencies.** Visualization of six attention heads from the qwen/Qwen2-0.5B-Instruct model. (Left) Each point represents a head in the positional–symbolic plane, labeled by its layer and head index. (Right) For the same heads, we plot their positional and symbolic scores as a function of RoPE frequencies.

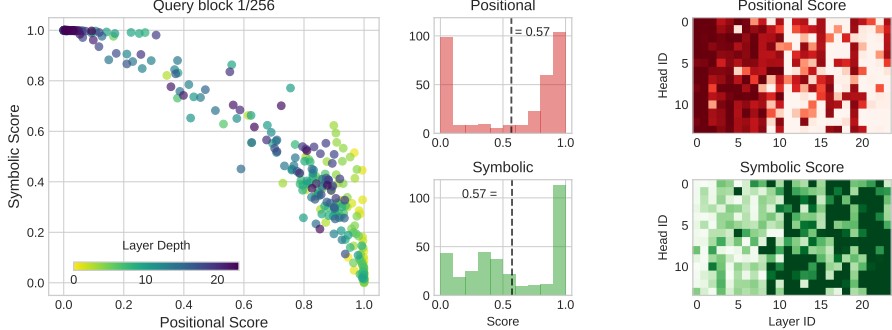

(b) **All heads, fixed first block.** (Left) Each head in the positional–symbolic plane. (Middle) Histogram showing predominance of symbolic over positional behavior. (Right) Heatmaps of positional and symbolic scores for each head across layers.

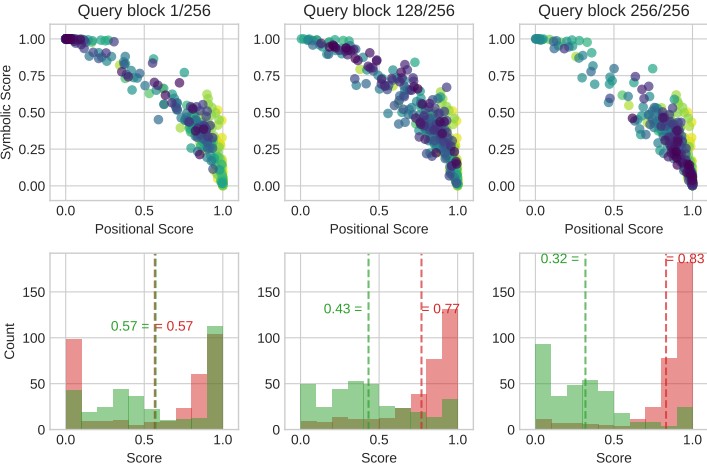

(c) **Heads when the block moves (1 to 9).** (Above) Heads in the positional–symbolic plane for each block position. (Bellow) Histograms showing how positional and symbolic scores change as the relevant block moves.

Figure 12: **Positional–Symbolic Plane across Heads.** 24 Layers, 14 Heads per layer. Since the model uses a head dimension of 64, this corresponds to 32 RoPE frequencies.

