# OpenReview forum: "Decoupling Positional and Symbolic Attention in Transformers"
_ICLR.cc/2026/Conference — ICLR 2026 Poster_

### Official Review · Reviewer_Fhmh · 2025-10-28

**Soundness:** 3
**Presentation:** 2
**Contribution:** 3
**Rating:** 8
**Confidence:** 4

**Summary:**

The authors formalise what it means for a head to behave positionally or symbolically. They then measure and study in detail both mathematically and empirically how positional and symbolical heads function. The paper focuses particularly on RoPE and performs interesting experiments to support their claims.

**Strengths:**

I found the paper to be insightful. The formalisation of positional / symbolic heads using invariance and equivariance is clever and the resulting analysis is very interesting. I found Section 5 to be a good sanity check of the theory and intuition.

I find this work to be a really strong and useful deep dive into RoPE consolidating and improving our existing understanding of how RoPE functions and how internal frequencies are used. As RoPE is the current most popular LLM PE, I find this paper to be on an important topic.

**Weaknesses:**

The main weakness I see is that the paper is dense at times especially in notation. It might be useful to add a small summary at the end of the sections to help convey what the authors believe should be the main message. This is mostly a minor suggestion to help with readability.

**Questions:**

I was wondering if the authors have an opinion given their analysis on the proposed method of p-RoPE (Barbero et al. 2024) where the p% lowest frequencies are set to spin to 0 to construct robust symbolic channels?

On a related note, do the authors have an idea of how the study could point towards possibly improving RoPE or for example the understanding of the choice of the wavelength hyperparameter. This is particularly important as many frontier models have been choosing different wavelengths for RoPE to deal with long context.

---

> ### Author Response · Authors · 2025-11-21
>
> We appreciate the reviewer’s positive assessment of our work and the time dedicated to formulating insightful comments and questions. Below, we address each questions raised by the reviewer:
>
> 1. “I was wondering if the authors have an opinion given their analysis on the proposed method of p-RoPE (Barbero et al. 2024) where the p% lowest frequencies are set to spin to 0 to construct robust symbolic channels?”
>
> Answer: Our results provide a mechanistic interpretation for why the p-RoPE method proposed by Barbero et al. (2024) generates better performance. Setting a fraction of the smallest frequencies to 0 allows indeed for positional information not to interrupt the desired symbolic behaviour. In fact, in our Figure 2A, we show that our symbolic (retrieval) task has maximal accuracy when angles are set to 0.
>
> 2. “On a related note, do the authors have an idea of how the study could point towards possibly improving RoPE or for example the understanding of the choice of the wavelength hyperparameter. This is particularly important as many frontier models have been choosing different wavelengths for RoPE to deal with long context.”
>
> Answer: Our work gives theoretical and empirical evidence for why choosing fixed wavelengths and frequencies eventually limits a model’s ability to handle positional and symbolic tasks simultaneously. Therefore, for a desired task-agnostic architecture, we believe it is necessary to explore more sophisticated mechanisms where hyperparameters are flexibly chosen during computation (perhaps sometimes deviating from perfect rotations), or even learnt during training.
>
> 3. “The main weakness I see is that the paper is dense at times especially in notation. It might be useful to add a small summary at the end of the sections to help convey what the authors believe should be the main message. This is mostly a minor suggestion to help with readability.”
>
> Answer: We thank the reviewer for the suggestion. Space permitted, we will try to add a “take away” message at each section for the revised version.

---

> ### Comment · Reviewer_Fhmh · 2025-11-27
>
> Thank you for the response! I have read the other rebuttals and I am happy to keep my score.

---

### Official Review · Reviewer_tJZ7 · 2025-10-30

**Soundness:** 3
**Presentation:** 2
**Contribution:** 3
**Rating:** 8
**Confidence:** 2

**Summary:**

This paper investigates how Transformers decouple positional and symbolic information in attention heads. It provides formal definitions for positional and symbolic behavior and a theorem showing they are mutually exclusive unless the attention pattern is uniform. The authors introduce a metric to quantify the amount of positional and symbolic information, using it to analyze RoPE models. The main finding is that low-frequency angles from RoPE are used for symbolic tasks, while high-frequency angles are used for positional tasks. This is demonstrated by training a toy model in canonical positional and symbolic tasks. Moreover, this metric is also applied to pre-trained models, showing that it can be used to decouple positional and symbolic behavior in attention heads.

**Strengths:**

- The main takeaway is very interesting: attention heads use RoPE frequency as a tool to implement either positional or symbolic functions.
- The paper has originality in providing a theoretical analysis of the problem.
- The choice of training toy models trained in canonical tasks is a clear way to test the positional and symbolic functions of attention heads.
- The experiments are done in multiple pre-trained LLMs, which shows how generalizable the proposed metric is.

**Weaknesses:**

- Some of the claims lack statistical tests:
    - The claim that early layers are positional and late layers are symbolic (Lines 268-269) is based on visual inspection, not a statistical test.
    - The "negatively correlated" claim (Lines 271-272) also needs a proper correlation test with a p-value.
- The theory is on logits, but the metric uses softmax weights. The paper doesn't discuss this switch. Softmax is a nonlinear function and has a relativistic behavior, ie, when analyzing a token $d$ attention to another token $s$, the logit value itself is not the only important component, but how large this logit is in comparison to the other tokens $j \leq d, j \neq s$. I wonder if this property would somehow impact the analysis.
- The usage of blocks is not motivated. Clearly, it is important for efficiency, but this is not discussed in the paper, which makes the transitions between paragraphs in Section 3.3 a bit rough to read. Block sizes were not discussed either.

**Questions:**

**Questions:**
1. What are the limitations of basing the theory on logits but the empirical metric on softmax weights?
2. Why use blocks? Is it just for efficiency? What's the impact of this approximation?
3. What is the model's accuracy on the binding task? Are you only analyzing instances where the model was correct?
4. Can you add statistical tests (e.g., correlation) for the claims in Lines 268-269 (layer depth) and Lines 271-272 (negative correlation)?

**Suggestions:**
- "Fine-Tuning Enhances Existing Mechanisms" (https://arxiv.org/abs/2402.14811) is a relevant paper that could be cited. They identify the circuit that solves entity tracking as having heads that handle positional information and others that handle content information.

---

> ### Author Response · Authors · 2025-11-21
>
> We thank the reviewer for her/his positive appraisal of our work, and the time invested in raising valuable comments and questions. Below we provide a point-by-point reply to the questions raised:
>
> 1. “What are the limitations of basing the theory on logits but the empirical metric on softmax weights?”
>
> Answer: This was a presentation decision. In fact, basing our definitions on softmax weights would have led to the same theoretical results. We decided to present them at the logit level only for the sake of simplicity and to ensure applicability to a broader class of logit-based attention mechanisms, while keeping the metric adapted to the models we consider. Note that if an attention head exhibits positional (or symbolic) behavior at the logit level, it will necessarily exhibit positional (or symbolic) behavior at the softmax level as well, but not viceversa.
>
> 2. “Why use blocks? Is it just for efficiency? What's the impact of this approximation?”
>
> Answer: Using blocks is primarily for efficiency, yes. In order to assess the impact of this approximation, we ran additional analyses to see how block sizes affect the score by refining each block into three smaller sub-blocks, thereby increasing the number of effective blocks, and reducing the size of each of them. Our results show that our metric is largely unaffected by this increase in resolution, especially for heads exhibiting a rather “pure” behavior (which is where our scores are the most informative). We say that a head is pure if it presents either strong positional or symbolic behavior. Formally, we define purity as the absolute difference between positional and symbolic scores, where 1 means purely positional or purely symbolic, and 0 means mixed. We compared the original score (blocks) with this refined version (sub-blocks), and examined how the mean absolute error between both scores changed as a function of a head’s purity. We found that the original score matches the refined score quite well. For example, in the top 5% of purity (average purity ~0.91), the error stays below ~0.02. For heads in the top 25% (purity 0.64), the error barely increases to ~0.086. We thank the reviewer for this observation, and we are going to add this robustness analysis of the score to the revised version.
>
> 3. “What is the model's accuracy on the binding task? Are you only analyzing instances where the model was correct?”
>
> Answer: The model’s accuracy on the binding task is ~0.96, ~0.90, and ~0.96, for positions 1, 128 and 256 respectively (each computed for 33 inputs). Thus, the overall accuracy for 99 inputs is 0.949. Our results are based on analyzing all instances, correct and incorrect.
>
> 4. “Can you add statistical tests (e.g., correlation) for the claims in Lines 268-269 (layer depth) and Lines 271-272 (negative correlation)”
>
> Answer: With respect to layer depth: We performed a median test between early (1-13) and late (14-26) layers to evaluate the statistical significance of our claim. For the positional score, we found a median score of 0.83 for early layers and 0.56 for late layers (p-value < 0.0001). For the symbolic score, we found the opposite pattern, a median score of 0.49 for early layers and 0.67 for late layers (p-value < 0.01).
> With respect to the negative correlation: Pearson’s correlation: -0.91,  p-value < 0.0001.
>
> 5. "Fine-Tuning Enhances Existing Mechanisms" (https://arxiv.org/abs/2402.14811) is a relevant paper that could be cited. They identify the circuit that solves entity tracking as having heads that handle positional information and others that handle content information.”
>
> Answer: Thank you for the suggestion. We will add it.

---

> > ### Comment · Reviewer_tJZ7 · 2025-11-27
> >
> > Thank you for all the answers. I will keep my score.

---

### Official Review · Reviewer_uiez · 2025-10-30

**Soundness:** 3
**Presentation:** 2
**Contribution:** 2
**Rating:** 4
**Confidence:** 4

**Summary:**

This work aims to study the positional and symbolic properties of the Rotary Positional Encodings (RoPE). To this end, the authors provide formal definitions of the two properties and introduce a metric to assess the extent to which a given head behaves positionally or symbolically on a given input, which enables analyzing the positional and symbolic behaviors of a given model at different levels of granularity, and also visualizing of the positional-symbolic profile of the model.

First, they apply the metric to the transformer-basd LLM, Gemma-2m,  and reveal that RoPE with a high frequency tends to be very positional while a low frequency corresponds to symbolic patterns.

Second, they introduce two tasks, Index and retrieval,  that can be solved by pure positional (symbolic) headsusing a 1–layer attention-only transformer. Empirical studies show that a head can only act positionally when it uses fairly large frequencies. If we force it to use lower frequencies, its performance drops.Similarly, attention heads can perform well on intrinsically symbolic tasks if given access to low frequencies.

**Strengths:**

* This work provides a formal definition of positional and symbolic behavior of attentions and introduces metrics to quantify the two scores for LLMs, which would be useful for following studies
* They provide both theoretical analyses and empirical studies to support their findings. Extensive experiments have been conducted, including multiple LLMs, studies across real LLMs and toy transformers.
* The presented tasks, index and retrieval, are interesting, which offer a doable way to study the functions of positional and symbolic properties

**Weaknesses:**

* Some key related papers are not discussed in the current manuscript, which strongly undermine the novelty and contribution of this work.
[1] and [2] have already discussed the separation of positional and semantic encodings. More importantly, [3] reveals the attention weights in transformers can be divided into two components: positional weights and semantic weights. It further introduces two properties, locality and symmetry, to measure the two behaviors. Basically, a high (or low) locality corresponds to positional (or symbolic) behaviors. In other words, the locality here can depict the positional-symbolic profile of a language model, i.e., a high locality is aligned with the RoPE with a high frequency. However, the authors did not discuss the main distinction of this work compared to prior studies, which undermines the contribution of this work.

* The presentation requires further improments. (1) This work aims to study and interpret the patterns of RoPE, but the authors do not provide a basic background in the preliminary section while they did present the background of the transformer architecture. I would suggest introducing the basic concepts of RoPE, e.g., what is the frequency. (2) I appreciate that they provide dense figures, but some figures are hard to understand. The lack of clear titles and axis texts lead to a bad readability (see questions below).

* They have conducted extensive theoretical analyses and empirical studies, but it is not easy for me to derive the main conclusions from this manuscript, and it is quite vague about how to use their findings to improve the current RoPE



[1] Constituency Parsing with a Self-Attentive Encoder. ACL 2018

[2] Rethinking Positional Encoding in Language Pre-training. ICLR 2021

[3] The Locality and Symmetry of Positional Encodings. EMNLP 2023 Findings

**Questions:**

* the key concept of frequency is not introduced in the preliminary, which hampers the understanding of the subsequent sections. I would suggest introducing the basic background of the RoPE in section 2.
* in line 145, *"its logit sequence is invariant under permutation of the key vectors."* I would like to confirm if the key vectors or input vectors (with regard to $x$) are shuffled? I asked this since self-attention uses the input vectors as key, query, and value. I am confused by performing the permutaion of only the key vectors.
* some figures are confusing. What are the score, frequency IDs, and the two line plots in Figure 1C? What is the number, e.g, 1->32, in Figure E? What are the three points in Figure F?
* In Figure 3A, why a low frequency results in a U-shaped acc patterns? Why an encoder with a low frequency can deal with indexes of the beginnig positions?

---

> ### Author Response · Authors · 2025-11-21
> **Part 1/2**
>
> We thank the reviewer for the positive assessment of our work and for the time dedicated to formulating constructive comments and questions. Below, we provide a point-by-point response to the weaknesses and questions raised:
>
> Reply to Weaknesses:
>
> 1. We thank the reviewer for sharing these papers. They do indeed discuss issues related to the topic adressed in our work. Paper [1] hard wires the separation of positional and symbolic (or content) information present in the input, by forcing this separation in the input embedding. Paper [2] incorporates particular projection matrices for positions and symbolic information independently, which are then aggregated as two different components of the logits function. In our work, we do not force any separation of positional or symbolic information in the models. Rather, we relate these qualities to the behavior of the logits function, with a particular interest in the context of RoPE as the state-of-the-art positional encoding. We found that, in decoder-only Transformers, there is an emergence of heads tending to either adopt positional or symbolic behavior (which is learnt and not forced in the architecture) suggesting that many underlying mechanisms can be decomposed into purely symbolic and purely positional ones. In this sense, our work provides strong theoretical and empirical reasons for the study of architectures with the capacity of learning separate positional and symbolic logit functions. Finally, paper [3] investigates attention patterns and defines interesting metrics of locality and symmetry of attention patterns in Transformers. In comparison to [3], instead of considering attention weight patterns as they stand, our metric analyzes how these patterns change when permuting the input vectors, thus capturing mechanistics properties of a different nature. In particular, our metric is in principle agnostic to how focused these patterns are, and for instance both symbolic and positional behavior are perfectly compatible with a focused attention. On top of developing this novel metric, we obtain several theoretical results related to the emergence of symbolic and positional head behaviors in RoPE (and potentially any logits-based model) and how this affects accuracy. Hence, our work provides substantive novel theoretical and empirical information, above and beyond what is being proposed in the papers mentioned by the reviewer. We thank the reviewer for mentioning these papers and we will add them to improve the clarity on the added value of our work.
>
> 2. As suggested by the reviewer, we will include information related to RoPE frequencies in the preliminaries, and clarify the figures in the main text and captions.
>
> 3. Our theoretical and empirical analyses are intended to shed light into the internal mechanisms implemented by Transformers when solving certain fundamental tasks. We find that, by decomposing these mechanisms into well defined positional and symbolic parts, one can describe their workings in a simple way (e.g. based on vectors alignments and their rotations, see Figure 3.B) and explain how and when different design choices affect the accuracy of the model, depending on the requirements of the task at hand (its degree of positional versus symbolic processing). Specifically, we show that the choice of available frequencies in RoPE critically determines the range of positional versus symbolic processing that the model will be able to implement. For example, our results imply that the magnitude of the smallest frequencies in RoPE may limit its ability to implement symbolic mechanisms, suggesting that turning off a fraction of the smallest frequencies could lead to improvements in its performance over symbolic related tasks, thus explaining for instance the success of  p-RopE [Barbero et al., 2024].

---

> ### Author Response · Authors · 2025-11-21
> **Part 2/2**
>
> Reply to Questions:
>
> 1. “the key concept of frequency is not introduced in the preliminary, which hampers the understanding of the subsequent sections. I would suggest introducing the basic background of the RoPE in section 2.”
>
> Answer: We thank the reviewer for this suggestion. The extra space available for the revision allows us to add and formalise the concept of frequency in the preliminaries.
>
> 2. “in line 145, "its logit sequence is invariant under permutation of the key vectors." I would like to confirm if the key vectors or input vectors (with regard to ) are shuffled? I asked this since self-attention uses the input vectors as key, query, and value. I am confused by performing the permutaion of only the key vectors.”
>
> Answer:  To explain why we only consider permutations of the key vectors, we need to fall back on how a given attention vector at position $i$ is computed. For simplicity, we will keep the RoPE matrix out of the computation. This does not affect the logic behind our explanation. The attention vector at position $i$ is given by:
>
> $a_{i}= \sum_{j=1}^{i} w_{i,j}V_j$, where $w_{i,j}=softmax(\left<Qx_{i},Kx_{j} \right>)$ are the attention weights.
>
> We are only interested in the attention weights involved in the previous formula, namely, $w_{i,1},w_{i,2},\dots, w_{i,i}$. Note that there is only one query vector involved: $q_i = Qx_{i}$,  while all the key vectors $k_j=Kx_j$ for $j=1\dots i$ are needed. Further note that the value vectors are not involved in the computation of these weights. So, there are only the key vectors to permute. Finally, we also note that permuting the input vectors $x_1, x_2, \dots, x_{i-1}$ themselves would lead to an equivalent definition.
>
>
> 3. “some figures are confusing. What are the score, frequency IDs, and the two line plots in Figure 1C? What is the number, e.g, 1->32, in Figure E? What are the three points in Figure F?”
>
> Answer:
>
> With respect to figure 1C: The scores of the red and green lines in figure 1C represent, respectively, the positional and symbolic score (as defined in appendix B.3). Frequency IDs represent RoPE angular frequencies. By convention, lower frequency IDs correspond to higher angular frequencies, and conversely (as defined in the caption of figure 1).
>
> With respect to figure 1E: “1 -> 32” represents a block permutation between block 1 and 32. The same logic is applied for other numbers (i.e., blocks).
>
> With respect to figure 1F: The three dots represent the pos/sym score for three logits function, respectively computed on three frequencies: 127/green, 83/red and 0/blue on the head (12:0).
>
> This information can be found in a combination of main text and figure captions. Given the additional space, and as a reply to the comment raised by the reviewer, we will clarify this information in the main text and captions.
>
> 4. “In Figure 3A, why a low frequency results in a U-shaped acc patterns? Why an encoder with a low frequency can deal with indexes of the beginnig positions?”
>
> Answer: The explanation for the accuracy shape as a function of the task is given in appendix D.2 (Proof of Theorem 7: U-Shape and Inverted U-Shape). We will explicitly add this information in the main text. However, the main intuition, say for the U-shaped acc. pattern, is quite simple. Consider our index task. The query vector has to encode the targeted position, in the sense of maximizing the attention at the suited position. In the simplest case, this is achieved by letting the logits (as a function of position) to follow the shape of a cosine function, suitably translated (this is controlled by the query vector) in order to be maximal at the desired position. For low frequencies, e.g. so that the cosine function completes less than half of a turn, it will be the case that when the targeted position (and thus where the logits have to be maximised) is at the beginning, the logits will become negative for positions near the middle or closer to the end, yielding a more pronounced bell attention shape after applying softmax. If the targeted position is near the middle instead, the logit will be maximum there but will likely remain positive for most positions in the input, yielding a more flat attention pattern after applying softmax. This difference translates in better accuracy for positions at the beginning (and a similar reasoning applies for positions at the end).

---

### Official Review · Reviewer_x2bi · 2025-11-01

**Soundness:** 3
**Presentation:** 3
**Contribution:** 2
**Rating:** 6
**Confidence:** 2

**Summary:**

Authors formalize definition of positional and symbolic heads, define metrics, canonical tasks, prove several theorems and perform analysis of several open weight and toy models.  Characterization of a model by its positional-symbolic profile is a potentially powerful tool to analyze how particular models behave on different tasks.

**Strengths:**

Theoretical Formalization: The authors provide formal definitions for when an attention head behaves purely positionally (depending only on position, regardless of token value) or purely symbolically (depending only on the token value, regardless of position) and prove several theorems.

Introduce a novel metric is introduced to quantify the extent to which any given attention head exhibits positional or symbolic behavior on a specific input. This metric allows for granular analysis, creating a positional-symbolic profile for the entire model, visualized in the positional-symbolic plane.

Applying the metric to real-world LLMs (GEMMA-2, QWEN-2, and LLAMA-3 families) reveals a sharp correspondence between RoPE frequencies and attention head behavior.

Very interesting finding about high frequencies in RoPE

**Weaknesses:**

Analysis done on limited set of tasks for real world LLMs ( binding task for real models)
It is not clear how reliable and computationally expensive metric in practice

**Questions:**

Is score input dependent? Could the head act symbolically on one sequence and positionally on another?
How the score is affected by block boundaries?
Are you considering only swaps but not all permutations of input sequence? How does this affect score?
How expensive is calculation of such scores in practice?

line 389. nit: Frecuency ?

---

> ### Author Response · Authors · 2025-11-21
>
> We appreciate the reviewer’s evaluation of our work and the time invested in providing constructive feedback and inquiries. Below, we address each of the noted weaknesses and questions individually.
>
> Reply to Weaknesses:
> 1. Analysis done on limited set of tasks for real world LLMs ( binding task for real models).
>
> Answer: Our choice for selecting the binding task was motivated by evidence showing that the underlying mechanisms needed to solve this task lie on symbolic and positional representations (https://arxiv.org/pdf/2310.17191), which serves as a challenging testing ground for the theory developed in the article. That being said, we agree with the reviewer that inspecting other real-world tasks in the lens of our method is a natural research direction.
>
> Reply to Questions:
> 1. “Is score input dependent? Could the head act symbolically on one sequence and positionally on another?”
>
> Answer: Yes. The quality of being positional or symbolic is assigned to a head acting on an input, not to a head on its own. This is because in principle, heads might need to implement different mechanisms for a successful performance. Thus, a challenging aspect of defining a suitable metric was (as the reviewer correctly points out) to make it input sensitive. Note that by averaging over a suitable set of inputs, one can obtain a score reflecting the behavior of the head at different levels of granularity. The relevance of this feature is for example illustrated by our "transitional scores” analysis (Figure 7 appendix B.5.). There, we parametrically modify the input by changing the question at the end so that the correct answer within the input moves from the beginning to the end of the input sequence, and observe how the head behavior gradually shifts from positional to symbolic (and vice versa, depending on the head).
>
> 2. “How the score is affected by block boundaries?”
>
> Answer: This is an excellent question. Our metric allows the user to choose the length of the blocks. Our theoretical definition of positional and symbolic behavior considers blocks of length one (individual tokens). However, for efficiency reasons, we chose in our case to work with blocks of length 6. On the one hand, this choice matches the structure of the binding task, whose binding pairs have length 6. On the other hand, we still have 256 blocks per input, so that our choice constitutes a good approximation to the maximal resolution. In order to empirically test this, we ran additional analyses to see how block boundaries affect the score by refining each block into three smaller sub-blocks. Our results show that our metric is largely unaffected by this increase in resolution, especially for heads exhibiting a rather “pure” behavior (which is where our scores are the most informative). We say that a head is pure if it presents either strong positional or symbolic behavior. Formally, we define purity as the absolute difference between positional and symbolic scores, where 1 means purely positional or purely  symbolic, and 0 means mixed. We compared the original score (blocks) with this refined version (sub-blocks), and examined how the mean absolute error between both scores changed as a function of a head’s purity. We found that the original score matches the refined score quite well. For example, in the top 5% (average purity ~0.91), the error stays below ~0.02. For heads in the top 25% (purity 0.64), the error barely increases to ~0.086. We thank the reviewer for this observation, and we are going to add this robustness analysis of the score to the CR version.
>
> 3. “Are you considering only swaps but not all permutations of input sequence? How does this affect score? How expensive is calculation of such scores in practice?”
>
> Answer: Note that any permutation on a set of blocks can be decomposed into a composition of swaps, and therefore there is no loss of generality in restricting to only swaps. The number of swaps is quadratic in the number of blocks. If $n$ is the number of blocks and the model has $H$ heads, then, for a given input, the cost of the calculation of the scores is ~$Hn^2$ plus the cost of one forward pass (to obtain all the key vectors).
>
> 4. Typo: line 389. nit: Frecuency
> Answer: Thank you for spotting this typo, we revised it.

---

### Meta-Review · Area_Chair_oCSd · 2026-01-13

**Summary:**

From the reviews and discussion, reviewers’ concerns center on (i) novelty positioning vs. prior interpretability work on separating positional vs semantic/structural components, (ii) clarity and completeness of presentation (especially RoPE “frequency” background and figure readability), and (iii) practical reliability/cost and scope of the proposed metric and empirical validation.

- Novelty and missing related work: Reviewer uiez argues that “key related papers are not discussed … [which] strongly undermine the novelty and contribution,” citing prior work on separation and metrics (“locality and symmetry”). This reviewer also notes the manuscript is “quite vague about how to use their findings to improve the current RoPE.”

- Presentation and interpretability gaps: Reviewer uiez highlights missing preliminaries (“the key concept of frequency is not introduced”) and “figures are hard to understand” due to missing titles/axes. Reviewer dyQR similarly flags figure readability issues (“subplots are missing captions … [and] the … subplot is missing indication on the legend”).

- Empirical rigor and alignment between theory vs. measurement: Reviewer tJZ7 requests statistical validation for central claims (“based on visual inspection, not a statistical test”; correlation claims need “p-value”) and questions the “switch” between “theory … on logits” and “metric uses softmax weights,” emphasizing softmax’s comparative nature.

- Practical scope and computational cost of the metric: Reviewer x2bi notes the analysis is on a “limited set of tasks” (for real LLMs) and asks whether the metric is “reliable and computationally expensive.” Related questions include input dependence, block choices, and permutation coverage.

These concerns are largely addressable through rebuttal clarifications and targeted additions to the revision, with the most substantive risk remaining around (a) whether novelty is sufficiently distinguished from the cited prior work for all readers, and (b) whether the metric’s practical cost and task coverage are convincingly demonstrated beyond the showcased settings.

**Reviewer Concerns:**

Concerns (partially) addressed by the rebuttal / discussion:

1. Novelty vs. related work and “what is new”: In response to Reviewer uiez’s claim that missing related work “strongly undermine[s] the novelty,” the authors explicitly differentiate their contribution: “we do not force any separation … Rather, we relate these qualities to the behavior of the logits function,” and contrast with [3] by emphasizing their metric measures changes “when permuting the input vectors.” They also commit to add the cited papers: “we will add them to improve the clarity on the added value.”

2. Missing RoPE frequency background and figure confusion: To uiez’s concern that “frequency is not introduced,” authors commit: “add and formalise the concept of frequency in the preliminaries,” and directly clarify figure semantics (e.g., “lower frequency IDs correspond to higher angular frequencies”). They also acknowledge the need to “clarify … in the main text and captions.”

3. Statistical tests for key empirical claims: In response to tJZ7’s request for statistical validation, the authors provide concrete tests and p-values: median test for early vs. late layers (“p-value < 0.0001” and “p-value < 0.01”) and correlation (“Pearson’s correlation: -0.91, p-value < 0.0001”). Reviewer tJZ7 confirms: “Thank you for all the answers. I will keep my score.”

4. Logits-level theory vs. softmax-level metric: Addressing “the theory is on logits, but the metric uses softmax weights,” authors state that using softmax for definitions “would have led to the same theoretical results,” and clarify the implication direction: logit-level positional/symbolic behavior implies softmax-level behavior (“but not viceversa”).

5. Block approximation and robustness: To both x2bi/tJZ7 concerns on blocks, authors justify blocks as “primarily for efficiency” and provide a robustness study: for “top 5% … purity ~0.91, the error stays below ~0.02,” and for “top 25% … error … ~0.086,” concluding the metric is “largely unaffected” by refined resolution.

6. Permutations vs swaps and computational cost: To x2bi’s questions, authors note “any permutation … can be decomposed … into swaps,” and quantify cost as “~ H n^2 plus the cost of one forward pass (to obtain all the key vectors).”

7. Binding-task accuracy and whether only correct instances are analyzed: Authors report binding accuracy “overall … 0.949” (with position-wise accuracies) and state analyses include “all instances, correct and incorrect.”

Concerns still outstanding / only partially resolved:

1. Breadth of task coverage for real-world LLM evaluation: Reviewer x2bi’s concern that analysis uses a “limited set of tasks” is only partially addressed; authors justify the binding task as a “challenging testing ground” but frame broader coverage as “a natural research direction.” This remains an open scope limitation.

2. Practical runtime/engineering feasibility at scale: While authors provide asymptotic complexity (“~ H n^2 … plus … one forward pass”) and robustness evidence, the concern that it may still be “computationally expensive … in practice” (Reviewer x2bi) remains partially open without concrete wall-clock profiling across large models/contexts.

3. Actionability for improving RoPE: Reviewer uiez noted it was “vague” how to use findings to improve RoPE. Authors do propose a concrete direction (“turning off a fraction of the smallest frequencies … could lead to improvements,” referencing p-RoPE), but this remains somewhat speculative absent broader validation beyond the presented settings.

**Reviewer Scores:**

- **Reviewer x2bi (Rating 6)**: The reviewer did not post a score update in the discussion. Given the rebuttal added (i) a concrete robustness analysis for block choice (“error stays below \~0.02” for high-purity heads) and (ii) an explicit cost estimate (“~ H n^2 plus … one forward pass”), I expect a modest upward revision to 8 is plausible, though not certain due to their stated uncertainty (“Confidence: 2 … Math/other details were not carefully checked”) and their stance (“would not mind if paper is rejected”).
  - **Expected score**: 6 or 8.

- **Reviewer uiez (Rating 4)**: The reviewer did not participate further. The rebuttal directly targets their main objections—novelty/related work (“we will add them … clarify … added value”), missing RoPE frequency preliminaries (“add and formalise the concept of frequency”), figure clarifications, and a clearer “how to improve RoPE” hook (linking to p-RoPE and “turning off … smallest frequencies”). I therefore estimate the score could move from 4 to 6 (borderline), but a larger increase is uncertain because the original concern framed the omission as “strongly undermine the novelty,” and this reviewer had relatively high confidence (“Confidence: 4”).
  - **Expected score**: 4 or 6.

- **Reviewer tJZ7 (Rating 8)**: No score change. The reviewer explicitly states: “Thank you for all the answers. I will keep my score.”
  - **Expected score**: 8.

- **Reviewer Fhmh (Rating 8)**: No score change. The reviewer explicitly states: “Thank you for the response! … I am happy to keep my score.”
  - **Expected score**: 8.

---

### Decision · Program_Chairs · 2026-01-26

Accept (Poster)